# Evaluation of agricultural products e-commerce logistics service capabilities in Heilongjiang Province based on entropy weight TOPSIS method

Wei Chen, Huan Chen *, Jianhui Yin, Weinan Sun

East University of Heilongjiang, Harbin, China

* 15779268409@163.com

## Abstract

This study aims to evaluate the current status and relative advantages of the logistics service capabilities of agricultural products e-commerce in Heilongjiang Province, identify its development shortcomings and propose optimization paths. In the context of the rapid development of agricultural products e-commerce, logistics service capabilities have become a key factor affecting regional agricultural competitiveness and e-commerce efficiency. However, current related studies mostly focus on the macro level and lack a systematic evaluation of regional and long-term dynamics. To fill this research gap, this paper constructs a logistics service capability evaluation index system covering four dimensions: logistics foundation, operational efficiency, service quality and information level, which is refined into 12 third-level indicators. Using the entropy weight-TOPSIS method, this paper conducts longitudinal trend analysis and horizontal regional comparison based on the time series data of Heilongjiang Province from 2013 to 2022 and the cross-sectional data of 31 provinces and cities across the country in 2022. The study found that the logistics service capabilities of agricultural products e-commerce in Heilongjiang Province have shown a steady upward trend, especially in regional linkage and cold chain system construction. However, compared with developed regions in the country, there are still obvious gaps in infrastructure, digital management and terminal distribution efficiency. The main constraints include insufficient cold chain coverage, unreasonable layout of urban and rural logistics networks, and low technical integration. The findings of this study reveal the key bottlenecks in improving regional e-commerce logistics service capabilities, and put forward policy recommendations for strengthening the cold chain logistics system, optimizing urban and rural logistics layout, and promoting digital transformation. The study not only provides a theoretical basis and practical guide for improving the efficiency of agricultural product e-commerce logistics in Heilongjiang Province, but also provides experience for other major agricultural production areas to explore high-quality development paths, which has important practical significance and policy value.

**Data availability statement:** All relevant data are within the manuscript and its Supporting Information files.

**Funding:** This study was supported by the following funding sources: 1. Heilongjiang Province Economic and Social Development Key Project: "Research on the Quality and Safety Supervision Mechanism and Implementation Path of Heilongjiang High-quality Agricultural Products Driven by Digital Economy" (Project No.: 23301); 2. Harbin Science and Technology Bureau Science and Technology Plan Project: "Research on the Efficiency Evaluation of Green Technology Innovation in Equipment Manufacturing Industry for Intelligent Manufacturing" (Project No.: ZC2023ZJ014007); 3. Heilongjiang Oriental University Scientific Research Innovation Team Building Project (Project No.: HDFKYTD202108); 4. National Social Science Fund Project: "Research on Identification and Breakthrough of Key Core Technologies in the IC Industry from the Perspective of 'Dual Chain Integration'" (Project No.: 23BGL076); 5. China Business Statistics Society Project: "Research on the Goals, Paths and Policies for High-Quality Development of the Cultural Tourism Industry Driven by Digitalization" (Project No.: 2024STY82); 6. China Business Statistics Society Planning Project: "Research on the Integration Development Mechanism and Path of Green Finance and Enterprise Innovation Capability under the 'Dual Carbon' Goal" (Project No.: 2024STY21); 7. China Business Statistics Society Planning Project: "Research on the Implicit Collaborative Innovation Mechanism and Coupling Effect of Scientific Research Project Teams under the Perspective of Intellectual Property Protection" (Project No.: 2024STY134). The funders had no role in study design, data collection and analysis, decision to publish, or preparation of the manuscript.

**Competing interests:** NO authors have competing interests Enter: The authors have declared that no competing interests exist.

## 1. Introduction

In recent years, with the rapid development of e-commerce, agricultural products e-commerce logistics services have become an important part of the modern logistics system, driving traditional logistics towards efficiency and intelligence [1]. The efficiency of the logistics system not only directly affects the market competitiveness of agricultural products, but also has a far-reaching impact on the sustainable development of the agricultural economy [2,3]. In the context of uneven regional economic development, how to effectively improve the logistics service capabilities of agricultural products has become a key issue that needs to be urgently addressed in major agricultural provinces such as Heilongjiang Province [4,5]. As an important agricultural province in China, Heilongjiang Province has strong advantages in agricultural product production, but there is still a large gap with developed coastal provinces in terms of logistics infrastructure construction, technological innovation and operation management level [6]. This gap leads to higher logistics costs and inefficient services for agricultural products during circulation, which in turn limits their competitiveness in the national market [7]. Therefore, it is particularly important to systematically evaluate the logistics service capabilities of Heilongjiang's agricultural products e-commerce and identify its shortcomings and potential [8,9].

This study aims to construct a comprehensive evaluation index system for the logistics service capabilities of Heilongjiang's agricultural products e-commerce, and conduct a comprehensive analysis from four dimensions: logistics transportation, technological innovation, operation management and economic development [10,11]. This paper uses the entropy weight-TOPSIS model to comprehensively evaluate the logistics service capabilities of agricultural products e-commerce in Heilongjiang Province, and compares and analyzes it with other provinces and cities across the country [12]. Through this analysis, the study provides a scientific basis for improving the logistics service capabilities of agricultural products e-commerce in Heilongjiang Province, and puts forward targeted policy recommendations [13,14]. The contribution of this study is: on the one hand, it constructs an evaluation index system for agricultural products e-commerce logistics service capabilities based on the actual situation in Heilongjiang Province; on the other hand, it introduces the entropy weight-TOPSIS method to conduct a comprehensive evaluation of capabilities, highlighting the analysis of regional differences, and making up for the lack of attention to local differences in existing research [15,16]. The research in this paper provides practical operational suggestions for improving the logistics service capabilities of agricultural products in Heilongjiang Province, and provides a reference for improving the logistics service capabilities of other major agricultural provinces [17].

Compared with previous studies, this paper has achieved breakthroughs and improvements in research perspective, indicator system, data processing and method application [18]. First, existing research on the logistics service capabilities of agricultural products e-commerce is mostly concentrated on the macro level, and static cross-sectional data are generally used for horizontal comparison, lacking a systematic description of regional dynamic evolution, especially in the research of

major agricultural production areas such as Heilongjiang [19]. This paper is based on the ten-year panel data of Heilongjiang Province from 2013 to 2022, combined with the cross-sectional data of 31 provinces and cities across the country in 2022, to construct a comprehensive evaluation system that integrates vertical and horizontal aspects. It not only dynamically evaluates the development trajectory of Heilongjiang Province, but also reveals its relative ranking and structural shortcomings in the country, significantly enhancing the timeliness and comparative value of the research. Second, this paper is more refined and in line with the actual situation of e-commerce logistics in the construction of the indicator system [20]. Compared with previous studies that focused on basic indicators such as transportation costs and delivery time, this paper takes logistics carrying capacity, technological innovation capability, operational capability and economic development capability as the first-level dimension, and refines them into 12 third-level indicators, covering the core elements of modern e-commerce logistics such as cold chain coverage, intelligent warehousing, and information system construction, which improves the comprehensiveness and scientificity of the evaluation. In terms of methods, this paper introduces the entropy weight-TOPSIS combination model to overcome the bias risk brought by the traditional subjective weighting method. At the same time, it handles missing values through multiple interpolation methods to ensure the continuity and reliability of panel data evaluation, which is better than the common simple mean method or deletion method [21]. In addition, this paper focuses on Heilongjiang, a typical region with weak logistics infrastructure and sparse population. Through the traceability analysis and horizontal comparison of the causes of logistics capacity fluctuations, it accurately identifies the key bottlenecks that restrict development, such as lagging cold chain system construction, insufficient digital capabilities, and low regional distribution efficiency. It puts forward more feasible policy recommendations, such as promoting the integration of "digital + logistics", building a regional collaborative distribution network, and developing a localized e-commerce platform, which significantly enhances the practical pertinence and application value of the research. Therefore, compared with previous studies, this paper has achieved significant innovations in research perspective, data depth, indicator structure and policy guidance, providing a more scientific, systematic and popularizable theoretical and practical path for improving the e-commerce logistics service capabilities of agricultural products in major agricultural production areas.

## 2. Construction of an evaluation index system for agricultural products e-commerce logistics service capabilities

The construction process of the agricultural product e-commerce logistics service capability evaluation index system is shown in Appendix 1. In Appendix 1, the selection basis, construction method and how to determine the weight of each evaluation index are introduced in detail. Readers can refer to the content in Appendix 1 to further understand the research framework and index system of this part.

## 3. Evaluation method of agricultural products e-commerce logistics service capability

This paper constructs an evaluation index system for the logistics service capabilities of agricultural products e-commerce, covering four dimensions: logistics carrying capacity, technological innovation capability, operational capability, and economic development capability [22,23]. Regarding operational capability and logistics carrying capacity, based on the summary of existing research literature, they have been moved to the appendix. Through the construction and evaluation of these index systems, it is intended to provide a scientific basis for improving the logistics service capabilities of agricultural products e-commerce in Heilongjiang Province.

This study uses a combination of entropy weight method and TOPSIS method to construct a method for comprehensively evaluating the e-commerce logistics service capabilities of agricultural products in Heilongjiang Province [24,25]. The entropy weight method is used to determine the weight of the evaluation index [26–28], while the TOPSIS rule is used to rank different solutions to select the best solution.. See Fig 14 in Appendix 2 for the specific flow chart.The following are the specific steps of this method:

## 3.1. Entropy weight method

The entropy weight method is based on the concept of information entropy and is widely used in multi-index decision analysis [29,30]. Its core idea is to reflect the importance of indicators by calculating the information entropy of each indicator [31–33]. The larger the entropy value, the smaller the information content of the indicator and the lower the contribution to the evaluation result; conversely, the smaller the entropy value, the greater the importance of the indicator in the evaluation [34–36].

The specific steps are as follows.

Standardization: Before calculating the entropy value, the original data is first standardized. Standardization can eliminate the unit differences between different indicators and make the data in the same dimension. This step ensures fairness between different indicators.

Calculate the information entropy of each indicator: The information entropy value of each indicator reflects the distribution of the indicator [37,38]. If the value distribution of an indicator is very concentrated (most samples have the same value), its information entropy value is small, which means that the indicator has a weak ability to distinguish the evaluation. On the contrary, if the value distribution of an indicator is more dispersed, the information entropy value is large, which means that the indicator can provide more information [39].

Determine the weight of the indicator: According to the information entropy value of each indicator, the weight of each indicator can be determined [40]. The smaller the information entropy of the indicator, the greater its weight in the evaluation. Conversely, the larger the information entropy of the indicator, the smaller its weight [41,42]. Through this process, we have obtained the importance ranking of each evaluation indicator.

## 3.2. TOPSIS method

TOPSIS method is a multi-index decision-making method based on ranking of ideal solutions and negative ideal solutions [43]. The core of this method is to rank each evaluation object by calculating the distance between it and the ideal solution and the negative ideal solution. The ideal solution represents the best state of all indicators, while the negative ideal solution represents the worst state of all indicators [44–46].

The steps of the TOPSIS method are as follows.

Construct a decision matrix: Organize each evaluation indicator and its corresponding sample data into a decision matrix [47]. Each row represents a sample (such as the logistics capacity of a province), and each column represents an indicator (such as logistics transportation capacity, technological innovation level, etc.).

Standardize the decision matrix: In order to eliminate the influence of different indicator dimensions, the decision matrix needs to be standardized [48]. The standardized data will make all indicators comparable and eliminate the differences in different dimensions and units [49].

Calculate the ideal solution and negative ideal solution: For each indicator, the ideal solution is the best value of the indicator among all samples (the maximum value for positive indicators and the minimum value for reverse indicators), and the negative ideal solution is the worst value. The ideal solution and the negative ideal solution represent the "optimal situation" and the "worst situation", respectively.

Calculate the distance between each sample and the ideal solution and the negative ideal solution: By calculating the distance between each sample and the ideal solution and the negative ideal solution, the relative closeness of the sample to the optimal state can be evaluated [50]. Samples that are closer to the ideal solution indicate that their overall performance is better; samples that are closer to the negative ideal solution indicate that their performance is worse.

Calculate relative closeness: Relative closeness is determined based on the ratio of the distance between the sample and the ideal solution and the negative ideal solution [51]. The higher the relative closeness, the closer the sample is to the ideal solution, that is, the better the performance [52].

### 3.3. Comprehensive evaluation

Through the combination of entropy weight method and TOPSIS method, this study can comprehensively evaluate the logistics service capacity of agricultural products e-commerce in Heilongjiang Province. The entropy weight method assigns weights to each indicator, reflecting the relative importance of each indicator in the overall evaluation; the TOPSIS rule gives the ranking of the optimal solution by calculating the relative distance between each solution and the ideal solution and the negative ideal solution.

The advantage of this combined method is that the entropy weight method can objectively determine the weight of the indicator and eliminate human bias, while the TOPSIS rule provides intuitive and effective ranking results by comparing the ideal solution and the negative ideal solution. Through this method, we can comprehensively evaluate the performance of Heilongjiang Province in terms of logistics carrying capacity, technological innovation, operation management and economic development capabilities, and then put forward practical suggestions for improvement.

## 4. Empirical analysis of e-commerce logistics service capabilities in Heilongjiang Province and other provinces and cities in China

The statistical data used in this study are from the official statistical platforms of the whole country and various provinces and cities. Specifically, the relevant materials in the "Statistical Yearbook" and "China Industrial Statistical Yearbook" are mainly referenced. These yearbooks are important statistical publications regularly issued by the national and local governments, covering detailed data in various economic, social, industrial and other fields. By collecting and analyzing these official statistical data, the accuracy and authority of the research results can be ensured, and reliable data support can also be provided for the research, helping to gain an in-depth understanding of the development status and trends of different regions and industries.

Combined with the characteristics and development trends of Heilongjiang Province's agricultural products e-commerce logistics service capabilities, the entropy weight-TOPSIS model is used to conduct a comprehensive and classified evaluation of Heilongjiang Province's e-commerce logistics service capabilities in the past 10 years from 2013 to 2022, and the entropy weight-TOPSIS model is used to evaluate the comprehensive e-commerce logistics service capabilities of various provinces and cities in my country in 2022.

### 4.1. Evaluation of the logistics service capability of agricultural products e-commerce in Heilongjiang Province based on entropy weight method

The evaluation results of the e-commerce logistics service capabilities of agricultural products in Heilongjiang Province based on the entropy weight method, the specific analysis process and related data are shown in Appendix 3. In the appendix, the calculation process of the weights of each indicator using the entropy weight method, as well as the application and analysis results of the evaluation model are shown in detail. Readers can refer to the content in Appendix 3 to further understand the research methods and data support of this part.

**4.1.1. Calculation based on entropy weight model.** This study uses the entropy weight model to comprehensively evaluate the e-commerce logistics service capabilities of Heilongjiang Province in the past 10 years (2013–2022). At the same time, we conducted a detailed assessment of the evaluation indicators of logistics carrying capacity, technological innovation capability, operational capability, and economic development capability.

**4.1.2 Analysis of calculation results based on entropy weight model.** This paper selects the data of the e-commerce logistics service capacity index system of Heilongjiang Province in the past 10 years (2013–2022) as a sample, and uses the entropy method to evaluate the dynamic development of the e-commerce logistics service capacity of agricultural products in Heilongjiang Province, and dynamically evaluates the e-commerce logistics service capacity of agricultural products in Heilongjiang Province. Since some data are missing, it is necessary to reasonably replace the data. This paper uses SPSS software to process and analyze the missing values, and uses the multiple interpolation

method to generate multiple interpolation data sets through multiple interpolations, and then analyzes each data set and merges the results. Each interpolation value is generated based on the distribution of other variables, thereby providing an estimate of the uncertainty of interpolation. The multiple interpolation method is more robust than the simple interpolation method, which can better capture the distribution and changes of the data, reduce the deviation caused by interpolation, and obtain alternative data to replace the missing values. The original data of the logistics carrying capacity index of the e-commerce logistics service capacity of agricultural products in Heilongjiang Province is shown in Table 1.

The first step is to standardize the original data. Since the indicators selected in this paper are all positive indicators, the original data are calculated using formula (4–2) and formula (4–3) to obtain the standardization of each indicator.

The second step is to normalize the data and then calculate the weight of the indicator value of the jth sample year $P_{ij}$ under the jth indicator $i$ according to formula (4–4). The P matrix is shown in Table 2

third step is to calculate the entropy value e and information utility value d of each item according to formulas (4-5), (4-6), (4-7), (4-8) and (4–9), and then calculate the weight w of each item. The calculation results are shown in Table 3 below:

Table 3, entropy value (e) and information utility value (d) are calculated in the following way:

Entropy value (e): Entropy value reflects the dispersion of each indicator, that is, the uncertainty of the data. The higher the entropy value, the more unstable or dispersed the data of the indicator is, and the smaller the contribution to the overall evaluation is.

**Table 1. Original data of Heilongjiang Province's agricultural products e-commerce logistics service capability and logistics carrying capacity indicators.**

|  | Mileage of expressways | Number of highway trucks | Postal business outlets |
|---|---|---|---|
| 2013 | 0.41 | 45.78 | 2645 |
| 2014 | 0.41 | 47.64 | 3156 |
| 2015 | 0.43 | 47.76 | 4348 |
| 2016 | 0.44 | 49.43 | 4714 |
| 2017 | 0.45 | 48.26 | 5818 |
| 2018 | 0.45 | 48.48 | 6155 |
| 2019 | 0.45 | 34.01 | 6638 |
| 2020 | 0.45 | 33.11 | 8391 |
| 2021 | 0.45 | 33.8 | 8584 |
| 2022 | 0.47 | 33.53 | 9419 |

**Table 2. The dimensionless results of the index data of logistics service capability and logistics carrying capacity of agricultural products e-commerce in Heilongjiang Province.**

| P Matrix | Mileage of expressways | Number of highway trucks | Postal business outlets |
|---|---|---|---|
| 20 13 | 0.0000194 | 0.1396888 | 0.0000203 |
| 20 14 | 0.0000194 | 0.1601956 | 0.0152908 |
| 20 15 | 0.0645137 | 0.1615186 | 0.0509595 |
| 20 16 | 0.0967705 | 0.1799306 | 0.0619115 |
| 20 17 | 0.1290272 | 0.1670312 | 0.0949469 |
| 2018 | 0.1290272 | 0.1694567 | 0.1050311 |
| 2019 | 0.1290272 | 0.0099226 | 0.1194841 |
| 2020 | 0.1290272 | 0.0000180 | 0.1719398 |
| 2021 | 0.1290272 | 0.0076073 | 0.1777150 |
| 2022 | 0.1935409 | 0.0046305 | 0.2027010 |

**Table 3. Weight table of the index system of logistics service capability and logistics carrying capacity of agricultural products e-commerce in Heilongjiang Province.**

| index | Entropy value e | Information utility value d | Weight w |
|---|---|---|---|
| Mileage of expressways | 0.88690 | 0.1131 0 | 0.27383 |
| Number of highway trucks | 0.81609 | 0.18391 | 0.44526 |
| Postal business outlets | 0.88 400 | 0.11603 | 0.28092 |

According to formulas (4–5) and (4–6), the entropy value of the mileage of expressways is 0.88690, indicating that the data of this indicator is relatively dispersed and the contribution is relatively small.

According to formulas (4–5) and (4–6), the entropy value of the number of trucks in operation on highways is 0.81609, indicating that the information of this indicator is also relatively dispersed, but not as good as the "mileage of expressways".

According to formulas (4–5) and (4–6), the entropy value of postal business outlets is 0.88400, showing a similar data dispersion as the "mileage of expressways".

Information utility value (d): The information utility value indicates the effectiveness of each indicator, which is usually calculated by $d = 1 - e$. It reflects the contribution of the indicator to the decision-making process. The higher the value, the more important the indicator plays in the evaluation.

The information utility value of the mileage of expressways is $d = 1 - 0.88690 = 0.1131$, indicating that the information contribution of this indicator is small.

The information utility value of the number of trucks in operation on highways is $d = 1 - 0.81609 = 0.18391$, indicating that the contribution of this indicator is large.

The information utility value of postal business outlets is $d = 1 - 0.88400 = 0.11603$, and the contribution value is also small, similar to the "mileage of expressways".

Multiply the weight of the jth indicator by $i$ the proximity value of the $X_{ij}$ jth evaluation indicator of the sample in the standardized matrix, and the resulting evaluation value is $y_{ij}$. Then add up all the rows to get the total score for the ith year, as shown in Table 4.

Based on the comprehensive scores of the logistics service capability and logistics carrying capacity indicators of Heilongjiang Province's agricultural products e-commerce during the period 2013–2022, the overall development trend of logistics carrying capacity in the past ten years was obtained, that is, the dynamic evaluation result trend chart of Heilongjiang Province's agricultural products e-commerce logistics carrying capacity, as shown in Fig 1.

As can be seen from Fig 1, the logistics carrying capacity score increased from 0.3457 to 0.7474 from 2013 to 2018, showing a significant upward trend overall, but it fell back significantly in 2019. As for the overall decline, by further tracing back the specific indicator data of logistics carrying capacity, it can be seen that the number of high-speed roads and postal business outlets are increasing year by year, while the number of road-operated trucks has decreased significantly in 2019, which has led to a significant decline in logistics carrying capacity. The impact of the number of road-operated trucks is so great, mainly because the size of road-operated trucks determines the amount of cargo flow, which in turn has a great impact on the logistics industry, and then affects the logistics service capabilities of e-commerce. Therefore, to improve the logistics service capabilities of agricultural products e-commerce in Heilongjiang Province, logistics carrying

**Table 4. Comprehensive score table of logistics service capability and logistics carrying capacity indicators of agricultural products e-commerce in Heilongjiang Province from 2013 to 2022.**

| years | 2013 | 2014 | 2015 | 2016 | 2017 | 2018 | 2019 | 2020 | 2021 | 2022 |
|---|---|---|---|---|---|---|---|---|---|---|
| Score | 0.3457 | 0.4176 | 0.5616 | 0.6680 | 0.7275 | 0.7474 | 0.3727 | 0.4209 | 0.4477 | 0.5662 |

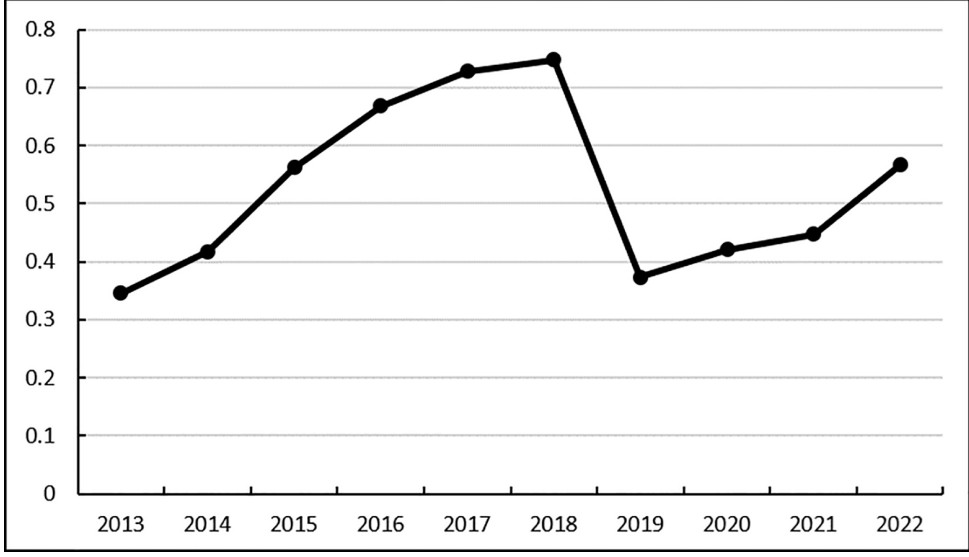

**Fig 1. Dynamic evaluation results of Heilongjiang Province's agricultural products e-commerce logistics service capability and logistics carrying capacity indicators.**

capacity is very important. At present, the number of road-operated trucks should be increased to improve logistics carrying capacity.

Similarly, a comprehensive evaluation of the technological innovation capabilities of the logistics service capabilities of agricultural products e-commerce in Heilongjiang Province during 2013–2022 was conducted. First, by collecting statistical data and supplementing missing data, the original data of technological innovation capability is obtained as shown in Table 5.

The first step is to standardize the original data. Since the indicators selected in this paper are all positive indicators, the original data are calculated using formula (4–2) and formula (4–3) to obtain the standardization of each indicator.

The second step is to normalize the data and then calculate the weight of the indicator value of the jth sample year $P_{ij}$ under the jth indicator $i$ according to formula (4–4). The P matrix is shown in Table 6

**Table 5. Original data of technical innovation capability indicators of agricultural product e-commerce logistics service capabilities in Heilongjiang Province.**

| | Mobile Internet users | Average number of mobile phones per 100 rural households at the end of the year |
|---|---|---|
| 20 13 | 1989.98 | 198.2 |
| 2014 | 2015.85 | 200.4 |
| 2015 | 2009.01 | 202.8 |
| 2016 | 2510.91 | 212.8 |
| 2017 | 2858.68 | 213.0 |
| 2018 | 2894.93 | 221.6 |
| 2019 | 2901.75 | 236.1 |
| 2020 | 2980.34 | 236.8 |
| 2021 | 3093.93 | 229.6 |
| 2022 | 3136.9 0 | 228.6 |

**Table 6. Dimensionless results of the index data of the logistics service capability and technological innovation capability of agricultural products e-commerce in Heilongjiang Province.**

| P Matrix | Mobile Internet users | Average number of mobile phones per 100 rural households at the end of the year |
|---|---|---|
| 2013 | 0.0000177 | 0.0000195 |
| 2014 | 0.0039845 | 0.0111165 |
| 2015 | 0.002931 0 | 0.0232436 |
| 2016 | 0.0802345 | 0.0737732 |
| 2017 | 0.1337986 | 0.0747838 |
| 2018 | 0.1393819 | 0.1182392 |
| 2019 | 0.1404323 | 0.1915071 |
| 2020 | 0.1525369 | 0.1950442 |
| 2021 | 0.1700322 | 0.1586629 |
| 2022 | 0.1766505 | 0.1536099 |

The third step is to calculate the entropy value e and information utility value d of each item according to formulas (4-5), (4-6), (4-7), (4-8) and (4–9), and then calculate the weight w of each item. The calculation results are shown in Table 7 below:

According to formula (4–5) and (4–6), the entropy value (e) of mobile Internet users is obtained: The entropy value of this indicator is 0.84926, which means that the data of this indicator is relatively scattered and the information is not concentrated. A higher entropy value means that the contribution of this indicator is smaller because its information uncertainty is higher. Information utility value (d): The information utility value is calculated by the formula $d = 1 - e$, that is, $d = 1 - 0.84926 = 0.15074$. This information utility value indicates that the effective information of this indicator is low and its contribution to decision-making is relatively small.

According to formula (4–5) and (4–6), the entropy value (e) of the average number of mobile phones owned by rural residents per 100 households at the end of the year is obtained: The entropy value of this indicator is 0.86491, which also means that the data dispersion of this indicator is large and the information uncertainty is high. Information utility value (d): The information utility value is calculated by the formula $d = 1 - e$, that is, $d = 1 - 0.86491 = 0.13509$. This information utility value indicates that the effective information of this indicator is low and its contribution to decision-making is relatively small.

Multiply the weight of the j-th indicator by $i$ the proximity value of the $X_{ij}$ j-th evaluation indicator of the sample in the standardized matrix, and the resulting evaluation value is $y_{ij}$. Then add up all the rows to get the total score for the i-th year, as shown in Table 8.

from Fig 2, Heilongjiang Province's technological innovation capability score increased from 0.0001 in 2013 to 0.9280 in 2018, and has been steadily increasing from 2013 to 2020, with no decline in the middle. The number of

**Table 7. Weight table of technical innovation capability index system for agricultural product e-commerce logistics service capabilities in Heilongjiang Province.**

| index | Entropy value e | Information utility value d | Weight w |
|---|---|---|---|
| Mobile Internet users | 0.84926 | 0.15074 | 0.5273 8 |
| Average number of mobile phones per 100 rural households at the end of the year | 0.86491 | 0.1350 9 | 0.4726 2 |

Table 7, entropy value (e) and information utility value (d) are calculated in the following way:

**Table 8. Comprehensive score table of technical innovation capability indicators of agricultural products e-commerce logistics service capability in Heilongjiang Province from 2013 to 2022.**

| years | 2013 | 2014 | 2015 | 2016 | 2017 | 2018 | 2019 | 2020 | 2021 | 2022 |
|-------|------|------|------|------|------|------|------|------|------|------|
| Score | 0.0001 | 0.0388 | 0.065 1 | 0.418 3 | 0.580 7 | 0.7026 | 0.8833 | 0.928 0 | 0.892 1 | 0.899 6 |

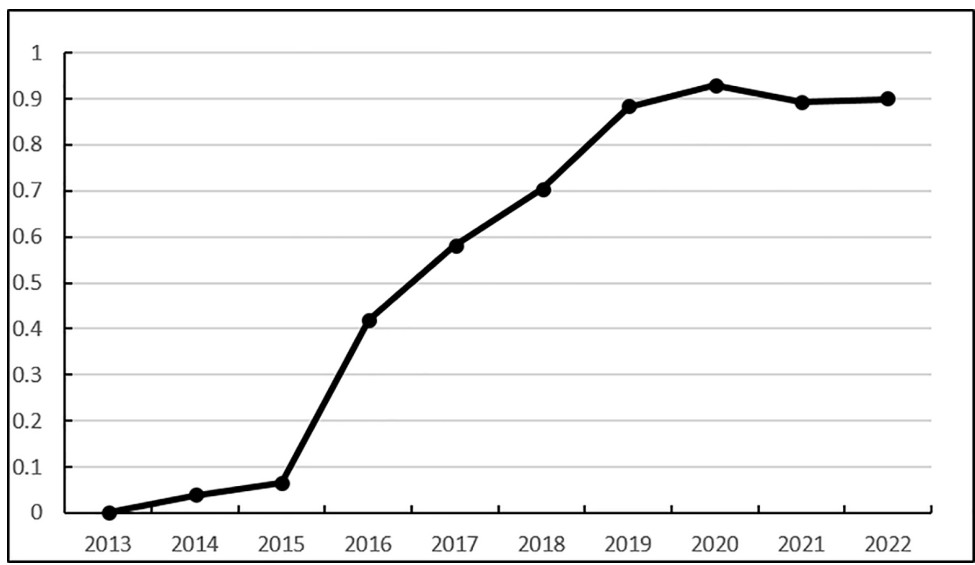

**Fig 2. Dynamic evaluation results of the technical innovation capability of Heilongjiang Province's agricultural products e-commerce logistics service capabilities.**

mobile Internet users has been increasing since 2013–2022, and the average number of mobile phones per 100 households in rural areas has been steadily increasing since 2013–2020, but it has decreased in 2021 compared with 2020. Phone ownership is a key factor in reflecting the convenience of rural residents' shopping. Therefore, in order to improve the logistics service capabilities of Heilongjiang Province's agricultural products e-commerce, it is necessary to increase investment in technological innovation and provide rural residents with more affordable mobile phones.

Similarly, a comprehensive evaluation is conducted on the operational capacity of agricultural product e-commerce logistics service capabilities in Heilongjiang Province during the period 20 13–20 22. First, the original data of operational capacity are obtained as shown in Table 9.

The first step is to standardize the original data. Since the indicators selected in this paper are all positive indicators, the original data are calculated using formula (4–2) and formula (4–3) to obtain the standardization of each indicator.

The second step is to normalize the data and then calculate the weight of the indicator value of the jth sample year $P_{ij}$ under the jth indicator $i$ according to formula (4–4). The P matrix is shown in Table 10

third step is to calculate the entropy value e and information utility value d of each item according to formulas (4-5), (4-6), (4–7), (4-8) and (4–9), and then calculate the weight w of each item. The calculation results are shown in Table 11 below:

According to formula (4–5) and (45–6), the entropy value (e) of express delivery volume is 0.837410, which means that the data dispersion of this indicator is large, the information uncertainty is high, and the contribution to the evaluation is small. Information utility value (d): The information utility value is calculated by formula d = 1 − e, and $d = 1 − 0.837410 = 0.162590$ is obtained, which means that the contribution of this indicator to the evaluation is small.

**Table 9. Original data of operational capability indicators of agricultural product e-commerce logistics service capabilities in Heilongjiang Province.**

|  | Express delivery volume | Agricultural product logistics turnover | Agricultural product logistics costs | Total agricultural product logistics | Value added of transportation, warehousing and postal services |
|---|---|---|---|---|---|
| 20 13 | 5393.85 | 972.92 | 1.1987 | 23.9 | 374.4 |
| 2014 | 7014.55 | 1008.46 | 1.1593 | 24.42 | 426.6 |
| 2015 | 12636.84 | 929.27 | 1.1742 | 27.74 | 434.3 |
| 2016 | 21769.77 | 904.76 | 1.1921 | 33.06 | 463.4 |
| 2017 | 23185.63 | 913.48 | 1.2243 | 41.77 | 488.7 |
| 2018 | 30177.15 | 810.66 | 1.2792 | 59.82 | 508.5 |
| 2019 | 35088.91 | 795.15 | 1.2919 | 88.09 | 503.6 |
| 2020 | 45522.34 | 694.04 | 1.3897 | 100.45 | 497.4 |
| 2021 | 60491 | 815.81 | 1.4564 | 120.12 | 528.1 |
| 2022 | 72638.07 | 846.09 | 1.5514 | 122.04 | 553.1 |

**Table 10. Dimensionless results of operational capability index data of agricultural product e-commerce logistics service capability in Heilongjiang Province.**

| P Matrix | Express delivery volume | Agricultural product logistics turnover | Agricultural product logistics costs | Total agricultural product logistics | Value added of transportation, warehousing and postal services |
|---|---|---|---|---|---|
| 20 13 | 0.0000259 | 0.1593353 | 0.0297529 | 0.0000244 | 0.0000173 |
| 20 14 | 0.0062338 | 0.1796407 | 0.0000296 | 0.0012922 | 0.0504778 |
| 20 15 | 0.0278591 | 0.1343963 | 0.0112518 | 0.0095423 | 0.0579238 |
| 2016 | 0.0629876 | 0.1203928 | 0.0247690 | 0.0227623 | 0.0860637 |
| 2017 | 0.0684335 | 0.1253748 | 0.0490848 | 0.0444064 | 0.1105290 |
| 2018 | 0.0953254 | 0.0666297 | 0.0905425 | 0.0892600 | 0.1296757 |
| 2019 | 0.1142178 | 0.0577682 | 0.1001330 | 0.1595100 | 0.1249374 |
| 2020 | 0.1543485 | 0.0000180 | 0.1739867 | 0.1902242 | 0.1189419 |
| 2021 | 0.2119233 | 0.0695721 | 0.2243552 | 0.2391036 | 0.1486291 |
| 2022 | 0.2586452 | 0.0868722 | 0.2960944 | 0.2438747 | 0.1728043 |

**Table 11. Weight table of the operational capability index system of agricultural product e-commerce logistics service capabilities in Heilongjiang Province.**

| index | Entropy value e | Information utility value d | Weight w |
|---|---|---|---|
| Express delivery volume | 0.837410 | 0.162590 | 0.221571 |
| Agricultural product logistics turnover | 0.924646 | 0.075354 | 0.102689 |
| Agricultural product logistics costs | 0.800306 | 0.199694 | 0.272134 |
| Total agricultural product logistics | 0.776545 | 0.223455 | 0.304515 |
| Value added of transportation, warehousing and postal services | 0.927286 | 0.072714 | 0.099091 |

Table 11, entropy value (e) and information utility value (d) are calculated in the following way:

According to formula (4–5) and (4–6), the entropy value (e) of agricultural product circulation volume is 0.924646. This value is large, indicating that the information of this indicator is not concentrated and the data changes greatly. Information utility value (d): The information utility value is calculated by formula $d = 1 - e$, and $d = 1 - 0.924646 = 0.075354$ is obtained, indicating that the contribution of this indicator to the evaluation is small and the information utility is low.

According to formulas (4–5) and (4–6), the entropy value (e) of agricultural product consumption is 0.800306, which means that the data dispersion of this indicator is large and the information uncertainty is high. Information utility value (d): The information utility value is calculated by the formula d = 1 − e, and $d$ = 1 − 0.800306 = 0.199694 is obtained, indicating that the contribution of this indicator to decision-making is medium.

According to formulas (4–5) and (4–6), the entropy value (e) of the total circulation of agricultural products is 0.776545, which means that the data dispersion of this indicator is small, the information is more concentrated, and the contribution to the evaluation is large. Information utility value (d): The information utility value is calculated by the formula d = 1 − e, and $d$ = 1 − 0.776545 = 0.223455 is obtained, indicating that the contribution of this indicator to the evaluation is large.

According to formulas (4–5) and (4–6), the entropy value (e) of the added value of transportation, warehousing and postal services is 0.927286, which is large, indicating that the data is highly dispersed and the uncertainty is high. Information utility value (d): The information utility value is calculated by the formula d = 1 − e, and the result is $d$ = 1 − 0.927286 = 0.072714, which means that the information utility of this indicator is low and its contribution to the evaluation is small.

Multiply the weight of the j-th indicator by $i$ the proximity value of the $X_{ij}$ j-th evaluation indicator of the sample in the standardized matrix, and the resulting evaluation value is $y_{ij}$. Then add up all the rows to get the total score for the i-th year, as shown in Table 12.

It is not difficult to see from Fig 3 that the operational capability score of Heilongjiang Province increased from 0.1185 in 2013 to 0.9470 in 2022. The operational capability of Heilongjiang Province 's agricultural product e-commerce logistics service capability has shown an overall upward trend in the past decade, and a slow upward trend during 2013–2017. Since 2018, the comprehensive operational capability score has shown a rapid upward trend. This is mainly because Heilongjiang Province has increased the establishment of express delivery outlets,

**Table 12. Comprehensive score table of operational capability indicators of agricultural product e-commerce logistics service capabilities in Heilongjiang Province from 2013 to 2022.**

| years | 2013 | 2014 | 2015 | 2016 | 2017 | 2018 | 2019 | 2020 | 2021 | 2022 |
|---|---|---|---|---|---|---|---|---|---|---|
| Score | 0.118 5 | 0.1386 | 0.156 2 | 0.2233 | 0.2942 | 0.3888 | 0.4937 | 0.5979 | 0.8113 | 0.9470 |

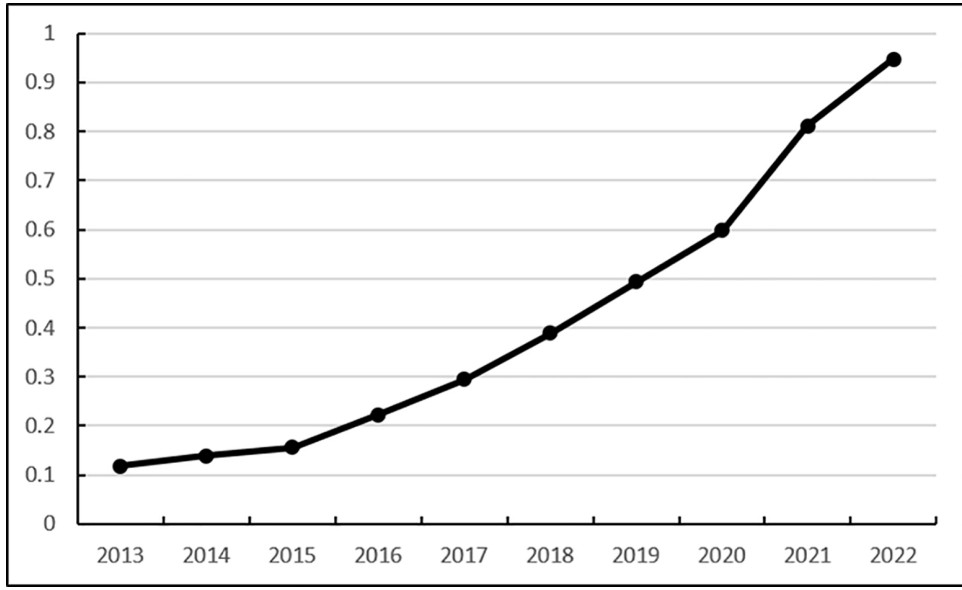

**Fig 3. Dynamic evaluation results of the operational capability of Heilongjiang Province's agricultural products e-commerce logistics service capabilities.**

which has led to a substantial increase in express delivery volume, increased total investment in agricultural product logistics, and gradually enhanced Heilongjiang Province's agricultural product e-commerce logistics service capability.

Similarly, a comprehensive evaluation is conducted on the economic development capacity of agricultural product e-commerce logistics service capacity in Heilongjiang Province during the period 20 13–20 22. First, the original data of economic development capacity are obtained as shown in Table 13.

The first step is to standardize the original data. Since the indicators selected in this paper are all positive indicators, the original data are calculated using formula (4–2) and formula (4–3) to obtain the standardization of each indicator.

The second step is to normalize the data and then calculate the weight of the indicator value of the jth sample year $P_{ij}$ under the jth indicator $i$ according to formula (4–4). The P matrix is shown in Table 14

The third step is to calculate the entropy value e and information utility value d of each item according to formulas (4–5), (4–6), (4–7), (4–8) and (4–9), and then calculate the weight w of each item. The calculation results are shown in Table 15 below:

Table 15, entropy value (e) and information utility value (d) are calculated in the following way:

According to formula (4–5) and (4–6), the entropy value (e) of total retail sales of consumer goods is 0.881633. A higher entropy value indicates that the data dispersion of this indicator is large, the information uncertainty is high, and the contribution is small. Information utility value (d): The information utility value is calculated by formula d = 1 − e, and d = 1 − 0.881633 = 0.118367 is obtained, indicating that the contribution of this indicator to decision-making is small.

**Table 13. Original data of economic development capability indicators of agricultural product e-commerce logistics service capabilities in Heilongjiang Province.**

|  | Total retail sales of consumer goods | Per capita disposable income of rural residents |
|---|---|---|
| 20 13 | 6251.2 | 9369 |
| 20 14 | 7015.3 | 10453 |
| 2015 | 4471 | 11095 |
| 2016 | 4794.1 | 11832 |
| 2017 | 5077.4 | 12665 |
| 2018 | 5275 | 13804 |
| 2019 | 5603.9 | 14982 |
| 2020 | 5092.3 | 16168 |
| 2021 | 5542.9 | 17889 |
| 2022 | 5210 | 18577 |

**Table 14. Dimensionless results of economic development capability index data of agricultural product e-commerce logistics service capability in Heilongjiang Province.**

| P Matrix | Total retail sales of consumer goods | Per capita disposable income of rural residents |
|---|---|---|
| 2013 | 0.1849875 | 0.0000213 |
| 2014 | 0.2643881 | 0.0251246 |
| 2015 | 0.0000264 | 0.0400047 |
| 2016 | 0.0335746 | 0.0570867 |
| 2017 | 0.0630134 | 0.0763937 |
| 2018 | 0.0835468 | 0.1027931 |
| 2019 | 0.117724 0 | 0.1300964 |
| 2020 | 0.0645617 | 0.1575852 |
| 2021 | 0.1113853 | 0.1974740 |
| 2022 | 0.0767924 | 0.2134203 |

**Table 15. Weight table of economic development capability index system of agricultural product e-commerce logistics service capability in Heilongjiang Province.**

| index | Entropy value e | Information utility value d | Weight w |
|---|---|---|---|
| Total retail sales of consumer goods | 0.881633 | 0.118368 | 0.492564 |
| Per capita disposable income of rural residents | 0.878059 | 0.121942 | 0.507436 |

According to formula (4–5) and (4–6), the entropy value (e) of per capita disposable income of rural residents is 0.878059, indicating that the data dispersion of this indicator is large and the information uncertainty is high. Information utility value (d): The information utility value is calculated by formula d = 1 − e, and $d = 1 - 0.878059 = 0.121941$ is obtained, indicating that the contribution of this indicator to the evaluation is large.

Multiply the weight of the jth indicator by $i$ the proximity value of the $X_{ij}$j- th evaluation indicator of the sample in the standardized matrix, and the evaluation value obtained is $y_{ij}$. Then add up all the rows to get the total score for the i-th year, as shown in Table 16.

As shown in Fig 4, the economic development capability score of Heilongjiang Province's agricultural product e-commerce logistics service capability has increased significantly from 0.3447 in 2013 to 0.5523 in 2014. However, it dropped to 0.0952 in 2015, and the score from 2016 to 2022 increased from 0.1983 to 0.6505, showing an overall upward trend. From the indicator level analysis, it can be seen that the per capita disposable income of rural residents showed

**Table 16. Comprehensive score table of economic development capability indicators of agricultural product e-commerce logistics service capabilities in Heilongjiang Province from 2013 to 2022.**

| years | 2013 | 2014 | 2015 | 2016 | 2017 | 2018 | 2019 | 2020 | 2021 | 2022 |
|---|---|---|---|---|---|---|---|---|---|---|
| Score | 0.3447 | 0.5523 | 0.0952 | 0.1983 | 0.2990 | 0.4001 | 0.5286 | 0.4950 | 0.6770 | 0.6505 |

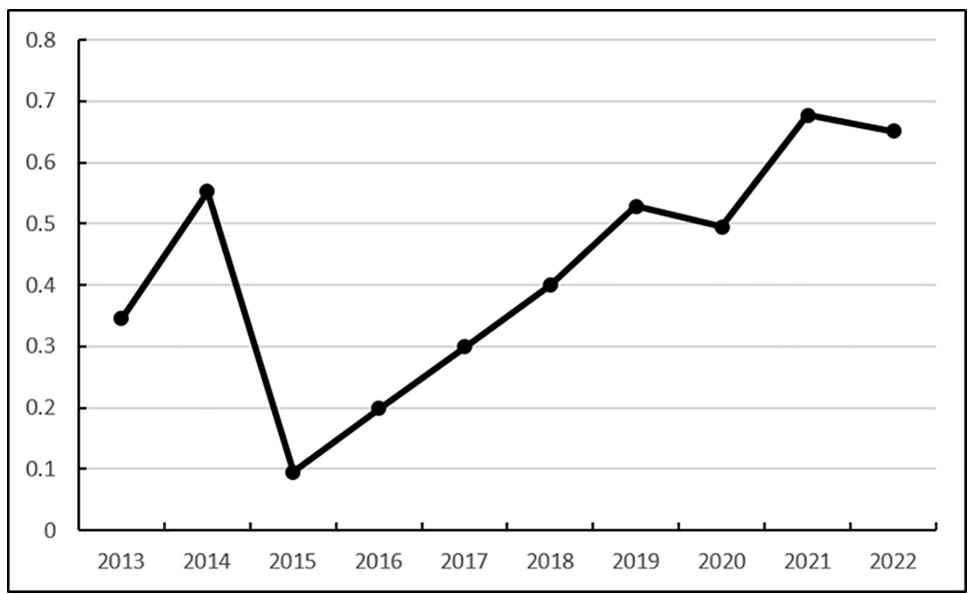

**Fig 4. Dynamic evaluation results of the economic development capability of Heilongjiang Province's agricultural products e-commerce logistics service capabilities.**

a stable growth trend from 2013 to 2022, while the total retail sales of consumer goods showed a fluctuating trend of increasing first, then decreasing, and then increasing again from 2013 to 2022, and the fluctuation trend was large, which also caused some fluctuations in the economic development capability indicator. Overall, Heilongjiang Province's economic development capability has maintained growth, which also provides basic support for the development of e-commerce logistics. The P matrix is shown in Table 17

to formulas (4-5), (4-6), (4-7), (4-8) and (4–9), the entropy value e and information utility value d of each item are calculated, and then the weight w of each item is calculated. The calculation results are shown in Table 18 below:

From the table, we can find that the entropy weights of logistics carrying capacity, technical level, operational capacity and economic development capacity are 0.2469, 0.1708, 0.4386 and 0.1436 respectively, that is, the importance of the secondary indicators from large to small is operational capacity, logistics carrying capacity, technical level and economic development capacity.

the jth indicator by the proximity value of the jth evaluation indicator of the ith sample in the standardized matrix to obtain the evaluation value of Xij as Yij. Then add up the rows to get the total score for the ith year, as shown in Table 19.

the comprehensive scores of Heilongjiang Province's agricultural products e-commerce logistics service capabilities during the period 2013–2022 in Table 19, the overall development trend of Heilongjiang Province's agricultural products e-commerce logistics service capabilities in the past decade was obtained, that is, the dynamic evaluation

**Table 17. Dimensionless results of the evaluation index data of agricultural product e-commerce logistics service capability in Heilongjiang Province.**

|  | Mileage of expressways | Number of highway trucks | Postal business outlets | Mobile Internet users | Average number of mobile phones per 100 rural households at the end of the year | Express delivery volume | Agricultural product logistics turnover |
|---|---|---|---|---|---|---|---|
| 2013 | 0.0000194 | 0.1396888 | 0.0000203 | 0.0000177 | 0.0000195 | 0.0000259 | 0.1593353 |
| 2014 | 0.0000194 | 0.1601956 | 0.0152908 | 0.0039845 | 0.0111165 | 0.0062338 | 0.1796407 |
| 2015 | 0.0645137 | 0.1615186 | 0.0509595 | 0.0029310 | 0.0232436 | 0.0278591 | 0.1343963 |
| 2016 | 0.0967705 | 0.1799306 | 0.0619115 | 0.0802345 | 0.0737732 | 0.0629876 | 0.1203928 |
| 2017 | 0.1290272 | 0.1670312 | 0.0949469 | 0.1337986 | 0.0747838 | 0.0684335 | 0.1253748 |
| 2018 | 0.1290272 | 0.1694567 | 0.1050311 | 0.1393819 | 0.1182392 | 0.0953254 | 0.0666297 |
| 2019 | 0.1290272 | 0.0099226 | 0.1194841 | 0.1404323 | 0.1915071 | 0.1142178 | 0.0577682 |
| 2020 | 0.1290272 | 0.000018 | 0.1719398 | 0.1525369 | 0.1950442 | 0.1543485 | 0.000018 |
| 2021 | 0.1290272 | 0.0076073 | 0.177715 | 0.1700322 | 0.1586629 | 0.2119233 | 0.0695721 |
| 2022 | 0.1935409 | 0.0046305 | 0.202701 | 0.1766505 | 0.1536099 | 0.2586452 | 0.0868722 |
|  | Agricultural product logistics costs | Total agricultural product logistics | Value added of transportation, warehousing and postal services | Total retail sales of consumer goods | Per capita disposable income of rural residents |  |  |
| 2013 | 0.0297529 | 0.0000244 | 0.0000173 | 0.1849875 | 0.0000213 |  |  |
| 2014 | 0.0000296 | 0.0012922 | 0.0504778 | 0.2643881 | 0.0251246 |  |  |
| 2015 | 0.0112518 | 0.0095423 | 0.0579238 | 0.0000264 | 0.0400047 |  |  |
| 2016 | 0.0247690 | 0.0227623 | 0.0860637 | 0.0335746 | 0.0570867 |  |  |
| 2017 | 0.0490848 | 0.0444064 | 0.1105290 | 0.0630134 | 0.0763937 |  |  |
| 2018 | 0.0905425 | 0.0892600 | 0.1296757 | 0.0835468 | 0.1027931 |  |  |
| 2019 | 0.1001330 | 0.1595100 | 0.1249374 | 0.1177240 | 0.1300964 |  |  |
| 2020 | 0.1739867 | 0.1902242 | 0.1189419 | 0.0645617 | 0.1575852 |  |  |
| 2021 | 0.2243552 | 0.2391036 | 0.1486291 | 0.1113853 | 0.1974740 |  |  |
| 2022 | 0.2960944 | 0.2438747 | 0.1728043 | 0.0767924 | 0.2134203 |  |  |

**Table 18. Entropy weight of evaluation index system of agricultural product e-commerce logistics service capability in Heilongjiang Province.**

| First level indicator | Secondary indicators | Level 3 indicators | Weight |
|---|---|---|---|
| Evaluation Index System of E -commerce Logistics Service Capabilities of Agricultural Products in Heilongjiang ProvinceA | Logistics carrying capacity B1 (0.2469) | Mileage of expressways (km) C1 | 0.0676 |
| | | Number of highway-operated trucks (10,000 vehicles) C2 | 0.1099 |
| | | Postal business outlets (offices) C3 | 0.0694 |
| | Technical Level B2 (0.1708) | Mobile Internet users (10,000) C4 | 0.0901 |
| | | Average number of mobile phones per 100 rural households at the end of the year (units) C5 | 0.0807 |
| | Operational Capabilities B3 (0.4386) | Express delivery volume (10,000 pieces) C 6 | 0.0972 |
| | | Agricultural product logistics turnover (100 million tons) C 7 | 0.0450 |
| | | Agricultural product logistics costs (trillion) C 8 | 0.1194 |
| | | Total agricultural product logistics (trillion) C 9 | 0.1336 |
| | | Added value of transportation, warehousing and postal services (100 million yuan) C 10 | 0.0435 |
| | Economic development capabilities B4 | Total retail sales of consumer goods (100 million yuan) C1 1 | 0.0708 |
| | | Per capita disposable income of rural residents (yuan) C1 2 | 0.0729 |

**Table 19. Comprehensive score of agricultural product e-commerce logistics service capabilities in Heilongjiang Province from 2013 to 2022.**

| years | 2013 | 2014 | 2015 | 2016 | 2017 | 2018 | 2019 | 2020 | 2021 | 2022 |
|---|---|---|---|---|---|---|---|---|---|---|
| Score | 0.1869 | 0.2499 | 0.2319 | 0.3628 | 0.4508 | 0.5326 | 0.5354 | 0.5958 | 0.7160 | 0.8023 |

result trend chart of Heilongjiang Province's agricultural products e-commerce logistics service capabilities, as shown in Fig 5.

It can be seen from Fig 5 that the comprehensive evaluation results of Heilongjiang Province's agricultural products e-commerce logistics service capabilities from 2013 to 2022 generally showed an upward trend. Heilongjiang Province's agricultural products e-commerce logistics service capabilities generally maintained growth from 2013 to 2018, but declined to a certain extent in 2019, which was due to the impact of the "mask-wearing period" that year. Its overall trend is stable and upward, indicating that Heilongjiang Province's logistics capabilities are constantly increasing and its service capabilities are constantly improving. Judging from the evaluation results of the classification indicators, the operational capacity and economic development capacity showed an overall continuous upward trend from 2013 to 2022, and the logistics carrying capacity and technological innovation capacity showed an overall upward trend from 2013 to 2022. However, due to the impact of the "mask-wearing period", it showed a trend of first decreasing and then increasing from 2018 to 2020. This also shows that the logistics carrying capacity construction and technological innovation capabilities of Heilongjiang Province's logistics industry are not stable enough, and when affected by the external environment, they are prone to large fluctuations. Heilongjiang Province needs to adjust these two aspects to improve the stability of logistics carrying capacity construction and technological innovation capabilities.

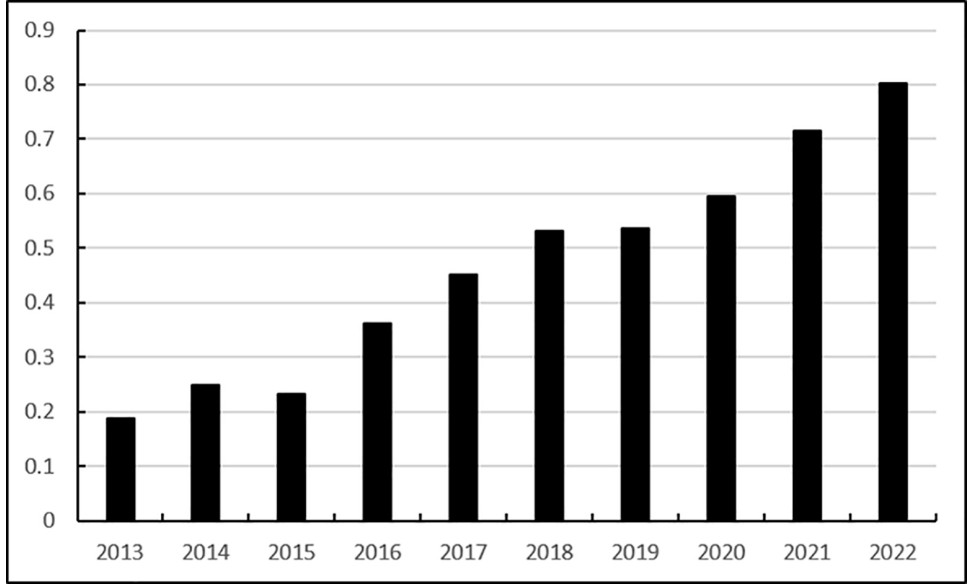

**Fig 5. Dynamic evaluation results of agricultural product e-commerce logistics service capabilities in Heilongjiang Province.**

## 4.2. Comprehensive evaluation of e-commerce logistics service capabilities in provinces and cities across the country based on entropy weight- TOPSIS method

### 4.2.1. Basic principles and calculation steps of TOPSIS method.

**4.2.1.1. *Basic principles of TOPSIS method.*** TOPSIS method, also known as the approach to ideal solution sorting method, is to sort by the distance between the best and worst solutions of the evaluation object. If the evaluation object is close to the best solution and far away from the worst solution, the score of the evaluation object is high, otherwise the score is low. The size of the proximity can reflect the quality of the evaluation object to a certain extent. The larger the $C\_I$ value, the higher the logistics service capacity of the agricultural products e-commerce in the region, and the smaller the $C\_I$ value, the lower the logistics service capacity of the agricultural products e-commerce in the region.

**4.2.1.2. *Calculation steps of TOPSIS method***

(1) Construct a weighted decision matrix:

$$v = (v_{ij})_{m \times n} \begin{bmatrix} w_1y_{11} & w_2y_{12} & \cdots & w_ny_{1n} \\ w_1y_{21} & w_2y_{22} & \cdots & w_ny_{2n} \\ \vdots & \vdots & \vdots & \vdots \\ w_1y_{m1} & w_2y_{m2} & \cdots & w_my_{mn} \end{bmatrix} \tag{4-10}$$

Where $w_j$ is the weight and $y_{ij}$ is the standardized data.

(2) Calculate the positive ideal solution and the negative ideal solution:

$$v^+ = \{max(v_{ij}) | i = 1, 2, 3, \cdots, m\} \tag{4-11}$$

$$v^- = \{min(v_{ij}) | i = 1, 2, 3, \cdots, m\} \tag{4-12}$$

(3) Calculate the distance:

$$D_i^+ = \sqrt{\sum_{j=1}^{n} \left(v_{ij} - v_j^+\right)^2} \tag{4–13}$$

$$D_i^- = \sqrt{\sum_{j=1}^{n} \left(v_{ij} - v_j^-\right)^2} \tag{4–14}$$

(4) Calculate proximity:

$$C_i = \frac{D_i^-}{D_i^+ + D_i^-} \tag{4–15}$$

**4.2.2. Calculation based on entropy weight – TOPSIS model.** We selected provinces and cities in China as evaluation objects, and selected data from 2022. Using the entropy weight-TOPSIS model, we first conducted a comprehensive evaluation of the comprehensive capabilities of agricultural product e-commerce logistics service capabilities in each province. Subsequently, we evaluated the capabilities of these 31 provinces and cities in four aspects: logistics carrying capacity, technological innovation capability, operational capability, and economic development capability.

**4.2.3. Analysis of calculation results based on entropy weight- TOPSIS model.**

**4.2.3.1. *Evaluation of agricultural product e-commerce logistics carrying capacity in various regions***

(1) Before using the TOPSIS method to calculate, we treat each region in 2022 as a sample alternative for analysis. According to Excel, the weighted decision matrix V is determined using formula (4–10); the weighted normalized matrix X is calculated by using the weights of the logistics carrying capacity indicators obtained by the entropy method in the previous section and the normalized matrix V; by obtaining the ideal solution $V^+$ and the negative ideal solution $V^-$, and calculating according to the Euclidean distance, the proximity is obtained as shown in Table 20 and Table 21.

The positive ideal value, negative ideal value and proximity of the agricultural product e-commerce logistics carrying capacity are calculated based on the obtained weighted decision matrix. Since the proximity and the scores of the alternatives (provinces) are consistent, the ranking of each province is obtained.

(2) Analysis of evaluation results

According to the results in Table 21a and Fig 6, we know that C21 has a higher score (closeness), followed by C5 and C9, and C8 and C27 have lower scores (closeness). Since the closeness is consistent with the direction of the alternatives, we can derive the ranking of the 31 alternatives (provinces) $A_i$ (i = 1,2…31): A21 > A5 > A9 > A15 > A10 > A25 > A30 > A1 > A6 > A22 > A29 > A16 > A13 > A17 > A23 > A7 > A18 > A3 > A28 > A11 > A31 > A4 > A14 > A24 > A20 > A26 > A2 > A19 > A8 > A27 From the ranking, we know that among the 31 provinces, Shandong Province has the highest score in the logistics carrying capacity of agricultural product e-commerce logistics service capabilities, followed by Guangdong Province and Hebei Province, and the lowest scores are Ningxia, Hainan and Tibet. In addition, according to the distance between the positive and negative ideal values of the scores of various provinces and cities, it can be intuitively seen that the positive and negative ideal value gaps in Zhejiang, Sichuan and other places are small, while the positive and negative ideal value gaps in Hainan, Ningxia and Tibet are large.

**Table 20. Weighted decision matrix X for agricultural product e-commerce logistics carrying capacity.**

| X | C 1 | C2 | C3 |
|---|---|---|---|
| A 1 | 0.09656 | 0.18684 | 0.13313 |
| A 2 | 0.01515 | 0.02722 | 0.03305 |
| A 3 | 0.10603 | 0.06067 | 0.09381 |
| A 4 | 0.10224 | 0.02765 | 0.04755 |
| A 5 | 0.20448 | 0.16870 | 0.37921 |
| A6 | 0.14958 | 0.10090 | 0.11514 |
| A7 | 0.14958 | 0.01870 | 0.11115 |
| A8 | 0.01893 | 0.00004 | 0.01300 |
| A9 | 0.14958 | 0.34761 | 0.14778 |
| A10 | 0.14390 | 0.23912 | 0.19631 |
| A11 | 0.08141 | 0.09390 | 0.07115 |
| A12 | 0.13632 | 0.09359 | 0.13312 |
| A13 | 0.13064 | 0.07387 | 0.12899 |
| A14 | 0.07573 | 0.06181 | 0.05254 |
| A15 | 0.08899 | 0.22039 | 0.25426 |
| A16 | 0.11928 | 0.11707 | 0.10356 |
| A17 | 0.07384 | 0.14728 | 0.08652 |
| A18 | 0.13822 | 0.07147 | 0.05016 |
| A 19 | 0.03219 | 0.03107 | 0.00693 |
| A 20 | 0.06437 | 0.00457 | 0.00549 |
| A 21 | 0.14390 | 0.41630 | 0.20584 |
| A 22 | 0.10414 | 0.16456 | 0.07124 |
| A 23 | 0.11928 | 0.07551 | 0.12555 |
| A 24 | 0.00947 | 0.07008 | 0.05042 |
| A 25 | 0.16662 | 0.13303 | 0.24858 |
| A26 | 0.01893 | 0.02586 | 0.04114 |
| A27 | 0.00002 | 0.00540 | 0.00004 |
| A28 | 0.13632 | 0.07986 | 0.02076 |
| A29 | 0.18555 | 0.06761 | 0.10475 |
| A30 | 0.09278 | 0.10745 | 0.30946 |
| A31 | 0.06816 | 0.06545 | 0.08535 |

As shown in Fig 7, from the perspective of the regional distribution of provinces, Shandong, Guangdong, Hebei, Jiangsu, and Henan are ranked in the top five in the agricultural products e-commerce logistics carrying capacity, reflecting the developed agricultural production and perfect logistics infrastructure. Shandong, as a major agricultural province, has a sound logistics system and strong e-commerce carrying capacity; Guangdong and Jiangsu, as developed coastal provinces, rely on strong industrial and commercial logistics to promote the development of agricultural products logistics; Hebei and Henan have improved their logistics capabilities by virtue of their geographical location and radiation effect on surrounding provinces. Central and western provinces such as Sichuan, Anhui, and Hubei rank high, thanks to the accelerated construction of logistics infrastructure in recent years. These provinces are not only important agricultural production bases, but also transportation hubs, playing an important role in the national agricultural products e-commerce logistics network. In contrast, western provinces such as Xinjiang, Gansu, and Tibet have relatively weak logistics capabilities due to their remote geographical location and inconvenient transportation. Although large cities such as Beijing, Shanghai, and Tianjin have developed economies, as consumer terminals, their agricultural products logistics

**Table 21. Ideal value, negative ideal value and proximity of agricultural products e-commerce logistics carrying capacity.**

| province | Distance from positive ideal value $d^+$ | Distance from negative ideal value $d^-$ | Relative proximity C | Ranking |
|---|---|---|---|---|
| Anhui | 0.3533 | 0.2489 | 0.4132 | 8 |
| Beijing | 0.5541 | 0.0454 | 0.0757 | 28 |
| Fujian | 0.4665 | 0.1540 | 0.2481 | 19 |
| Gansu | 0.5211 | 0.1161 | 0.1822 | 23 |
| Guangdong | 0.2476 | 0.4626 | 0.6514 | 2 |
| Guangxi | 0.4150 | 0.2140 | 0.3402 | 10 |
| Guizhou | 0.4827 | 0.1872 | 0.2795 | 17 |
| Hainan | 0.5846 | 0.0229 | 0.0377 | 30 |
| Hebei | 0.2476 | 0.4062 | 0.6213 | 3 |
| Henan | 0.2618 | 0.3411 | 0.5658 | 5 |
| Heilongjiang | 0.4626 | 0.1431 | 0.2363 | 21 |
| Hubei | 0.4115 | 0.2122 | 0.3402 | 9 |
| Hunan | 0.4305 | 0.1978 | 0.3149 | 14 |
| Jilin | 0.4990 | 0.1109 | 0.1819 | 24 |
| Jiangsu | 0.2595 | 0.3480 | 0.5728 | 4 |
| Jiangxi | 0.4157 | 0.1966 | 0.3211 | 13 |
| Liaoning | 0.4185 | 0.1860 | 0.3078 | 15 |
| Inner Mongolia | 0.4812 | 0.1634 | 0.2535 | 18 |
| Ningxia | 0.5627 | 0.0452 | 0.0744 | 29 |
| Qinghai | 0.5734 | 0.0647 | 0.1015 | 26 |
| Shandong | 0.1837 | 0.4861 | 0.7258 | 1 |
| Shanxi | 0.4102 | 0.2073 | 0.3357 | 11 |
| Shaanxi | 0.4333 | 0.1889 | 0.3036 | 16 |
| Shanghai | 0.5158 | 0.0868 | 0.1440 | 25 |
| Sichuan | 0.3142 | 0.3274 | 0.5103 | 6 |
| Tianjin | 0.5488 | 0.0521 | 0.0867 | 27 |
| Tibet | 0.5953 | 0.0054 | 0.0089 | 31 |
| Xinjiang | 0.4963 | 0.1593 | 0.2430 | 20 |
| Yunnan | 0.4442 | 0.2235 | 0.3347 | 12 |
| Zhejiang | 0.3358 | 0.3404 | 0.5034 | 7 |
| Chongqing | 0.4775 | 0.1273 | 0.2104 | 22 |

scores are low and they rely on external supply. High land and logistics costs also limit their role in agricultural products logistics. In recent years, central and western provinces such as Shanxi, Yunnan, and Shaanxi have gradually improved their agricultural products logistics carrying capacity through policy support and infrastructure construction. Overall, the improvement of logistics capabilities depends on the improvement of infrastructure, the diversity of agricultural products and the promotion of market demand.

**4.2.3.2. *Evaluation of agricultural product e-commerce technology innovation capabilities in various regions***

(1) Before using the TOPSIS method to calculate, we treat each region in 2022 as a sample alternative.

So as to conduct analysis. According to Excel, the weighted decision matrix V is determined by formula (4–10); the weighted normalized matrix X is calculated by the weights of the logistics carrying capacity indicators obtained by the entropy method in the previous section and the normalized matrix V; by obtaining the ideal solution $V^+$ and the negative ideal solution $V^-$, and calculating according to the Euclidean distance, the proximity is obtained as shown in Table 22 and Table 23.

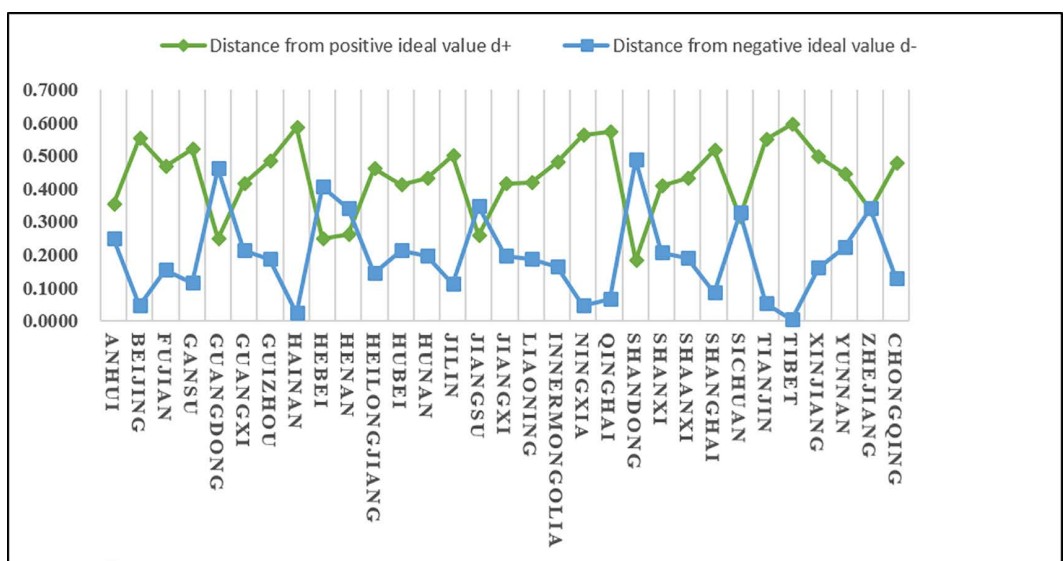

**Fig 6. The distance between the logistics carrying capacity of agricultural products e-commerce logistics service capabilities and the positive and negative ideal values of each province and city.**

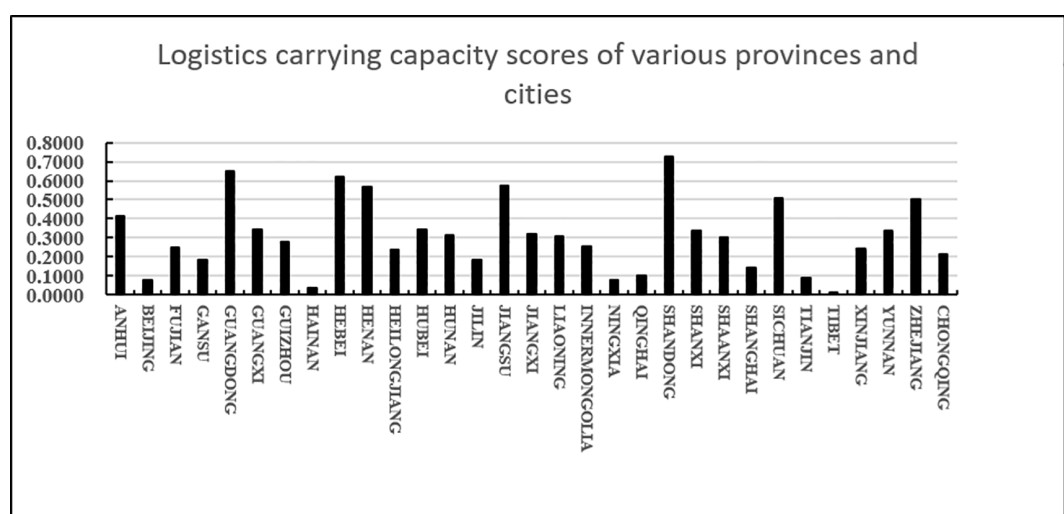

**Fig 7. Logistics carrying capacity scores of various provinces and cities.**

According to the obtained weighted decision matrix, the positive ideal value, negative ideal value and proximity of agricultural product e-commerce technology innovation capability are calculated. Since the proximity and the scores of the alternatives (provinces) are consistent, the ranking of each province is obtained.

(2) Analysis of evaluation results

According to the results in Table 23 and Fig 8, we know that C5 has a higher score (closeness), followed by C10 and C21, and C26 and C18 have lower scores (closeness). Since the closeness is consistent with the direction of

**Table 22. Weighted decision matrix of agricultural product e-commerce technology innovation capability X.**

| X | C 4 | C 5 |
|---|---|---|
| A 1 | 0.2360 | 0.2003 |
| A 2 | 0.1352 | 0.1714 |
| A 3 | 0.1827 | 0.1630 |
| A 4 | 0.1002 | 0.2814 |
| A 5 | 0.6791 | 0.2536 |
| A 6 | 0.2238 | 0.2261 |
| A 7 | 0.1704 | 0.3209 |
| A 8 | 0.0341 | 0.2599 |
| A 9 | 0.3340 | 0.1095 |
| A10 | 0.4176 | 0.2135 |
| A11 | 0.1301 | 0.0596 |
| A12 | 0.2246 | 0.1951 |
| A13 | 0.2687 | 0.2450 |
| A14 | 0.0977 | 0.1040 |
| A15 | 0.4010 | 0.1192 |
| A16 | 0.1686 | 0.2106 |
| A17 | 0.1910 | 0.0579 |
| A 18 | 0.1045 | 0.0863 |
| A 19 | 0.0223 | 0.2315 |
| A 20 | 0.0160 | 0.2553 |
| A 21 | 0.4346 | 0.0785 |
| A 22 | 0.1451 | 0.0799 |
| A 23 | 0.1774 | 0.1656 |
| A 24 | 0.1551 | 0.0000 |
| A 25 | 0.3717 | 0.1699 |
| A 26 | 0.0583 | 0.1252 |
| A 27 | 0.0001 | 0.1762 |
| A 28 | 0.0980 | 0.1115 |
| A 29 | 0.1824 | 0.2777 |
| A 30 | 0.3393 | 0.1206 |
| A 31 | 0.1444 | 0.1490 |

the alternatives, we can derive the ranking of the thirty-one alternatives (provinces) (i = 1,2…31): A5 > A10 > A21 > A15 > A25 > A30 > A13 > A9 > A7 > A6 > A1 > A29 > A12 > A16 > A4 > A3 > A23 > A8 > A2 > A20 > A31 > A17 > A19 > A22 > A27 > A24 > A28 > A11 > A14 > A18 > A26 From the ranking, we know that among the 31 provinces, Guangdong scored the highest in terms of agricultural product e-commerce technology innovation capability, followed by Henan and Shandong, and the lowest scores were in Tianjin and Inner Mongolia. In addition, according to the distance between the positive and negative ideal values of the scores of various provinces and cities, it can be intuitively seen that the positive and negative ideal value gaps in Zhejiang, Sichuan, Shandong and other places are relatively small, while the positive and negative ideal value gaps in Guangdong, Hainan, Inner Mongolia, Tibet, Tianjin and other places are relatively large.

shown in Fig 9, Guangdong, Henan, Shandong, Jiangsu, Sichuan and other provinces rank at the top in terms of agricultural product e-commerce logistics innovation technology capabilities. As economically developed coastal provinces,

**Table 23. Ideal value, negative ideal value and proximity of agricultural product e-commerce technology innovation capability.**

| province | Distance from positive ideal value$d^+$ | Distance from negative ideal value$d^-$ | Relative proximity C | Ranking |
|---|---|---|---|---|
| Anhui | 0.4591 | 0.0958 | 0.4027 | 11 |
| Beijing | 0.5641 | 0.0476 | 0.2789 | 19 |
| Fujian | 0.5209 | 0.0599 | 0.3197 | 16 |
| Gansu | 0.5803 | 0.0892 | 0.3398 | 15 |
| Guangdong | 0.0673 | 0.5253 | 0.9150 | 1 |
| Guangxi | 0.4650 | 0.1012 | 0.4062 | 10 |
| Guizhou | 0.5087 | 0.1320 | 0.4166 | 9 |
| Hainan | 0.6478 | 0.0687 | 0.2880 | 18 |
| Hebei | 0.4047 | 0.1235 | 0.4648 | 8 |
| Henan | 0.2827 | 0.2199 | 0.6239 | 2 |
| Heilongjiang | 0.6079 | 0.0205 | 0.1905 | 28 |
| Hubei | 0.4715 | 0.0885 | 0.3868 | 13 |
| Hunan | 0.4173 | 0.1322 | 0.4656 | 7 |
| Jilin | 0.6205 | 0.0203 | 0.1869 | 29 |
| Jiangsu | 0.3435 | 0.1750 | 0.5491 | 4 |
| Jiangxi | 0.5222 | 0.0727 | 0.3406 | 14 |
| Liaoning | 0.5544 | 0.0398 | 0.2646 | 22 |
| Inner Mongolia | 0.6207 | 0.0183 | 0.1791 | 30 |
| Ningxia | 0.6628 | 0.0541 | 0.2598 | 23 |
| Qinghai | 0.6663 | 0.0654 | 0.2774 | 20 |
| Shandong | 0.3443 | 0.1950 | 0.5619 | 3 |
| Shanxi | 0.5858 | 0.0274 | 0.2204 | 24 |
| Shaanxi | 0.5251 | 0.0589 | 0.3160 | 17 |
| Shanghai | 0.6144 | 0.0240 | 0.2015 | 26 |
| Sichuan | 0.3424 | 0.1670 | 0.5441 | 5 |
| Tianjin | 0.6509 | 0.0191 | 0.1750 | 31 |
| Tibet | 0.6942 | 0.0310 | 0.2024 | 25 |
| Xinjiang | 0.6177 | 0.0220 | 0.1936 | 27 |
| Yunnan | 0.4986 | 0.1103 | 0.3998 | 12 |
| Zhejiang | 0.3944 | 0.1296 | 0.4772 | 6 |
| Chongqing | 0.5616 | 0.0430 | 0.2697 | 21 |

Guangdong and Jiangsu have promoted continuous innovation in technologies such as intelligent warehousing and automation by relying on sound logistics infrastructure and e-commerce markets. Zhejiang has also achieved remarkable results in logistics technology innovation by relying on e-commerce giants such as Alibaba. As transportation hubs in central China, Henan and Sichuan have rapidly improved their logistics innovation capabilities through policy support and technology investment. Midstream provinces such as Guizhou, Guangxi, and Anhui have improved their logistics innovation technology levels through e-commerce poverty alleviation and smart logistics technology. Large cities such as Beijing, Shanghai, and Tianjin have weak agricultural product logistics innovation capabilities because they rely more on external supply chains. Western and northeastern regions such as Tibet, Xinjiang, and Heilongjiang have relatively slow logistics technology innovation due to geographical and infrastructure limitations. Although Qinghai and Ningxia are not ranked high, with national policies and the construction of the "Belt and Road Initiative", the logistics innovation capabilities of the western region are expected to be further improved.

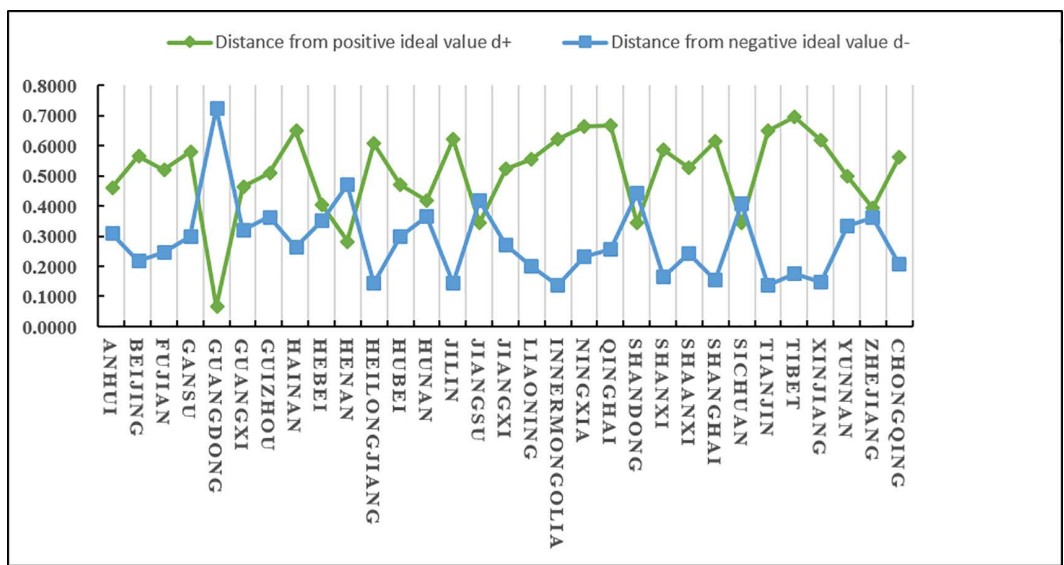

**Fig 8. The distance between the technological innovation capability of agricultural product e-commerce and the positive and negative ideal values of each province and city.**

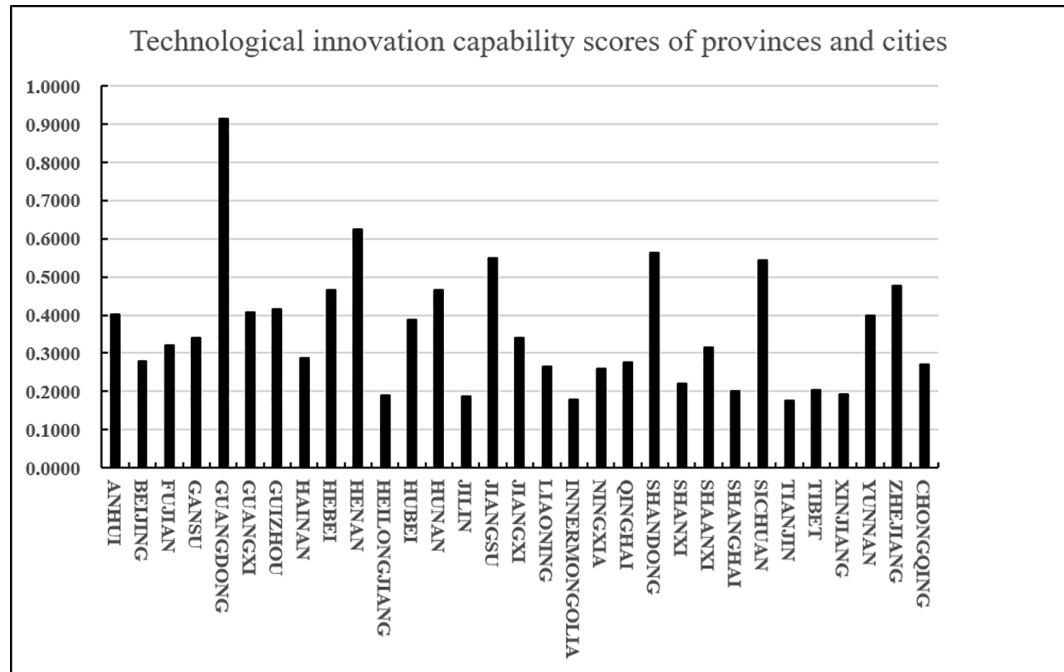

**Fig 9. Technological innovation capability scores of provinces and cities.**

### 4.2.3.3. *Evaluation of agricultural product e-commerce operation capabilities in various regions*

(1) Before using the TOPSIS method to calculate, we treat each region in 2022 as a sample alternative for analysis. According to Excel, the weighted decision matrix V is determined using formula (4–10); the weighted normalized matrix X is calculated by using the weights of the logistics carrying capacity indicators obtained by the entropy method in the previous section and the normalized matrix V; by obtaining the ideal solution $V^+$ and the negative ideal solution $V^-$, and calculating according to the Euclidean distance, the proximity is obtained as shown in Table 24 and Table 25.

According to the obtained weighted decision matrix, the positive ideal value, negative ideal value and proximity of agricultural product e-commerce technology innovation capability are calculated. Since the proximity and the scores of the alternatives (provinces) are consistent, the ranking of each province is obtained.

**Table 24. Weighted decision matrix X of agricultural product e-commerce technology innovation capability.**

| X | C 6 | C 7 | C 8 | C 9 | C 10 |
|---|---|---|---|---|---|
| A 1 | 0.0370 | 0.0187 | 0.0187 | 0.0227 | 0.0444 |
| A 2 | 0.0204 | 0.0260 | 0.0260 | 0.1147 | 0.0171 |
| A 3 | 0.0446 | 0.0289 | 0.0289 | 0.0923 | 0.0408 |
| A4 | 0.0019 | 0.0094 | 0.0094 | 0.0125 | 0.0106 |
| A5 | 0.3163 | 0.1318 | 0.1318 | 0.1912 | 0.0856 |
| A6 | 0.0109 | 0.0249 | 0.0249 | 0.0216 | 0.0219 |
| A7 | 0.0050 | 0.0283 | 0.0283 | 0.0089 | 0.0162 |
| A8 | 0.0016 | 0.0014 | 0.0014 | 0.0234 | 0.0071 |
| A9 | 0.0552 | 0.1580 | 0.1580 | 0.0271 | 0.0621 |
| A10 | 0.0466 | 0.0799 | 0.0799 | 0.0239 | 0.0778 |
| A11 | 0.0075 | 0.0083 | 0.0083 | 0.0106 | 0.0110 |
| A12 | 0.0336 | 0.0312 | 0.0312 | 0.0247 | 0.0480 |
| A13 | 0.0242 | 0.0496 | 0.0496 | 0.0141 | 0.0348 |
| A14 | 0.0060 | 0.0047 | 0.0047 | 0.0042 | 0.0112 |
| A15 | 0.0914 | 0.1662 | 0.1662 | 0.1389 | 0.0739 |
| A16 | 0.0190 | 0.0363 | 0.0363 | 0.0138 | 0.0265 |
| A17 | 0.0179 | 0.0503 | 0.0503 | 0.0312 | 0.0270 |
| A18 | 0.0024 | 0.0062 | 0.0062 | 0.0096 | 0.0262 |
| A19 | 0.0009 | 0.0104 | 0.0104 | 0.0017 | 0.0036 |
| A20 | 0.0002 | 0.0000 | 0.0000 | 0.0016 | 0.0024 |
| A21 | 0.0605 | 0.1934 | 0.1934 | 0.0993 | 0.0953 |
| A22 | 0.0073 | 0.0018 | 0.0018 | 0.0280 | 0.0234 |
| A 23 | 0.0117 | 0.0271 | 0.0271 | 0.0291 | 0.0253 |
| A 24 | 0.0299 | 0.0234 | 0.0234 | 0.2015 | 0.0390 |
| A 25 | 0.0300 | 0.0399 | 0.0399 | 0.0304 | 0.0326 |
| A 26 | 0.0126 | 0.0207 | 0.0207 | 0.0625 | 0.0204 |
| A 27 | 0.0000 | 0.0274 | 0.0274 | 0.0000 | 0.0000 |
| A 28 | 0.0016 | 0.0277 | 0.0277 | 0.0216 | 0.0169 |
| A 29 | 0.0092 | 0.0032 | 0.0032 | 0.0167 | 0.0271 |
| A 30 | 0.2404 | 0.1628 | 0.1628 | 0.1829 | 0.0471 |
| A 31 | 0.0113 | 0.0288 | 0.0288 | 0.0212 | 0.0218 |

**Table 25. Ideal value, negative ideal value and proximity of agricultural product e-commerce operation capability.**

| province | Distance from positive ideal value$d^+$ | Distance from negative ideal value$d^-$ | Relative proximity C | Ranking |
|---|---|---|---|---|
| Anhui | 0.4167 | 0.0045 | 0.1392 | 15 |
| Beijing | 0.3966 | 0.0152 | 0.2371 | 9 |
| Fujian | 0.3780 | 0.0138 | 0.2373 | 8 |
| Gansu | 0.4577 | 0.0004 | 0.0441 | 27 |
| Guangdong | 0.0883 | 0.1787 | 0.8272 | 1 |
| Guangxi | 0.4333 | 0.0023 | 0.0998 | 19 |
| Guizhou | 0.4414 | 0.0020 | 0.0913 | 21 |
| Hainan | 0.4607 | 0.0006 | 0.0507 | 26 |
| Hebei | 0.3197 | 0.0576 | 0.4287 | 5 |
| Henan | 0.3611 | 0.0215 | 0.2890 | 7 |
| Heilongjiang | 0.4555 | 0.0004 | 0.0432 | 28 |
| Hubei | 0.4075 | 0.0060 | 0.1596 | 13 |
| Hunan | 0.4068 | 0.0069 | 0.1697 | 11 |
| Jilin | 0.4621 | 0.0002 | 0.0312 | 30 |
| Jiangsu | 0.2377 | 0.0883 | 0.5556 | 3 |
| Jiangxi | 0.4216 | 0.0039 | 0.1287 | 16 |
| Liaoning | 0.4047 | 0.0071 | 0.1721 | 10 |
| Inner Mongolia | 0.4585 | 0.0009 | 0.0601 | 25 |
| Ningxia | 0.4635 | 0.0002 | 0.0317 | 29 |
| Qinghai | 0.4726 | 0.0000 | 0.0061 | 31 |
| Shandong | 0.2755 | 0.0974 | 0.5311 | 4 |
| Shanxi | 0.4519 | 0.0014 | 0.0762 | 23 |
| Shaanxi | 0.4275 | 0.0031 | 0.1149 | 17 |
| Shanghai | 0.3782 | 0.0441 | 0.3571 | 6 |
| Sichuan | 0.4029 | 0.0061 | 0.1621 | 12 |
| Tianjin | 0.4205 | 0.0053 | 0.1479 | 14 |
| Tibet | 0.4526 | 0.0015 | 0.0788 | 22 |
| Xinjiang | 0.4387 | 0.0023 | 0.0983 | 20 |
| Yunnan | 0.4533 | 0.0011 | 0.0687 | 24 |
| Zhejiang | 0.1016 | 0.1464 | 0.7903 | 2 |
| Chongqing | 0.4302 | 0.0027 | 0.1080 | 18 |

(2) Analysis of evaluation results

According to the results in Table 25 and Fig 10, we know that C5 has a higher score (closeness), followed by C30 and C15, and C14 and C20 have a lower score (closeness). Since the closeness is consistent with the direction of the alternatives, we can derive the ranking of the 31 alternatives (provinces) A$_i$ (i = 1,2…31): A5 > A30 > A15 > A21 > A9 > A24 > A10 > A3 > A2 > A17 > A13 > A25 > A12 > A26 > A1 > A16 > A23 > A31 > A6 > A28 > A7 > A27 > A22 > A29 > A18 > A8- > A4 > A11 > A19 > A14 > A20 From the ranking, we know that among the 31 provinces, Guangdong scored the highest in terms of agricultural product e-commerce operational capability, followed by Zhejiang and Jiangsu, and the lowest scores were in Jilin and Qinghai. In addition, according to the distance between the positive and negative ideal values of the scores of various provinces and cities, it can be intuitively seen that the positive and negative ideal value gaps in Jiangsu, Shandong and other places are relatively small, while the positive and negative ideal value gaps in Guangdong, Gansu, Inner Mongolia, Ningxia, Xinjiang and other places are relatively large.

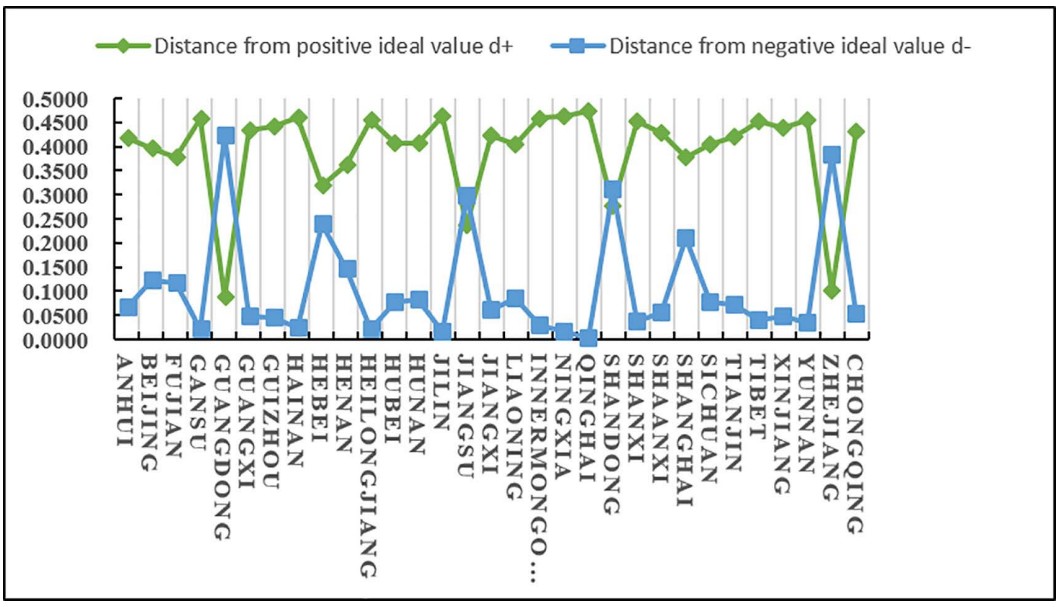

**Fig 10. The distance between the agricultural products e-commerce operation capacity of each province and city and the positive and negative ideal values.**

It is easy to see from Fig 11 that developed coastal provinces such as Guangdong, Zhejiang, Jiangsu, Shandong, and Hebei are in a leading position in agricultural product e-commerce operations. They mainly rely on their complete e-commerce infrastructure and efficient logistics network to promote the supply of agricultural products. The process of digitalization and intelligence of the chain. Jiangsu and Shandong are particularly outstanding in terms of operational capabilities due to their rich agricultural resources and their close integration with the logistics system. Central and southwestern regions such as Henan, Hunan, and Sichuan have significantly improved their e-commerce operation capabilities through policy support and the gradual improvement of logistics infrastructure. The rise of these regions shows that the potential of the central

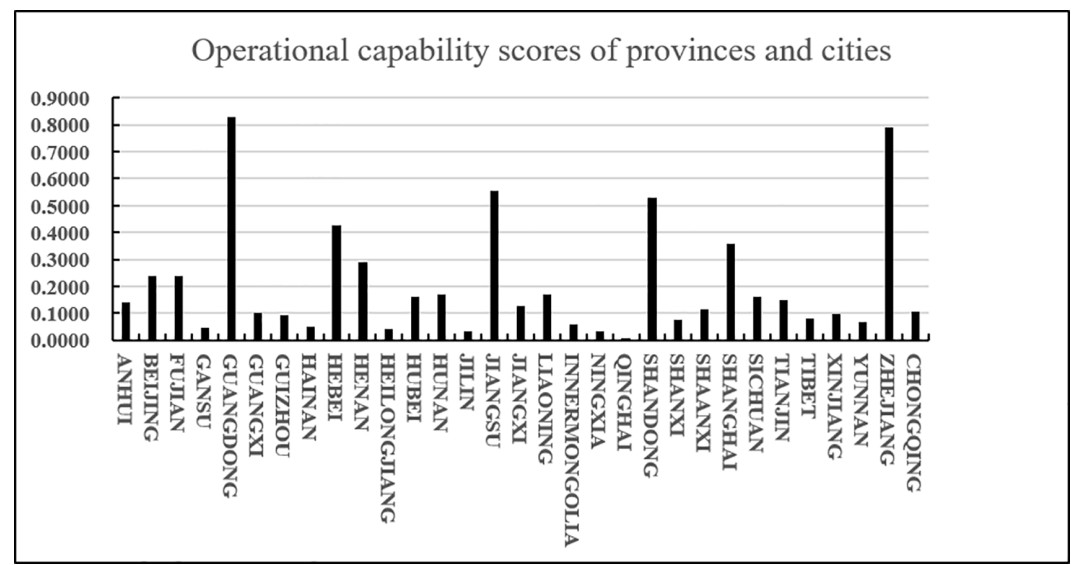

**Fig 11. E-commerce operation capability scores of various provinces and cities.**

and western regions in the field of agricultural products e-commerce is gradually released. In contrast, although provinces such as Fujian, Beijing, Liaoning, and Hubei have relatively complete e-commerce logistics systems, there is a certain gap between them and the leading provinces in terms of technological innovation and logistics network coverage. The western and northeastern provinces, such as Xinjiang, Guizhou, Tibet, Gansu, etc., are restricted by their geographical conditions, relatively backward logistics infrastructure and small market size, and their agricultural product e-commerce operation capabilities are relatively limited. However, with the continued support of national policies and the advancement of the "One Belt, One Road" initiative, the e-commerce logistics network and operational capabilities in these regions are expected to be significantly improved in the future. Overall, coastal provinces continue to lead by virtue of their comprehensive advantages, the central and western regions are showing a rising trend, while the western and northeastern regions are expected to further enhance their e-commerce operation capabilities through policy guidance and technological progress.

4.2.3.4. ***Evaluation of the agricultural product economic development capacity of each region***

(1) Before using the TOPSIS method to calculate, we treat each region in 2022 as a sample alternative for analysis. According to Excel, the weighted decision matrix V is determined using formula (4–10); the weighted normalized matrix X is calculated using the weights of the logistics carrying capacity indicators obtained by the entropy method in the previous section and the normalized matrix V; by obtaining the ideal solution $V^+$ and the negative ideal solution $V^-$, and calculating according to the Euclidean distance, the proximity is obtained as shown in Table 26 and Table 27.

According to the obtained weighted decision matrix, the positive ideal value, negative ideal value and proximity of agricultural product e-commerce technology innovation capability are calculated. Since the proximity and the scores of the alternatives (provinces) are consistent, the ranking of each province is obtained.

(2) Analysis of evaluation results

According to the results in Table 27 and Fig 12, we know that C15 has a higher score (closeness), followed by C30 and C5, and C4 and C20 have a lower score (closeness). Since the closeness is consistent with the direction of the alternatives, we can derive the ranking of the 31 alternatives (provinces) $A_i$ (i = 1,2…31): A15 > A30 > A5 > A21 > A24 > A2 > A3 > A10 > A25 > A12 > A1 > A13 > A26 > A31 > A9 > A16 > A17 > A29 > A23 > A6 > A18 > A22 > A11 > A7 > A8 > A14 > A27 > A28 > A19 > A4 > A20 From the ranking, we know that among the 31 provinces, Jiangsu scored the highest in terms of agricultural product e-commerce operational capability, followed by Zhejiang and Guangdong, and Gansu and Qinghai scored the lowest. In addition, based on the distance between the positive and negative ideal values of the scores of various provinces and cities, it can be intuitively seen that the gap between the positive and negative ideal values of Beijing and Shanghai is relatively small, while the gap between the positive and negative ideal values of Gansu, Guizhou, Ningxia, Xinjiang and other places is relatively large.

As can be seen from Fig 13, coastal provinces such as Jiangsu, Zhejiang, Guangdong, and Shandong have significant advantages in the field of agricultural product e-commerce with their developed economic foundations, complete e-commerce infrastructure and logistics networks. These provinces are rich in agricultural resources. They have realized the digitalization and market expansion of agricultural production through e-commerce platforms, and their branding strategies have further enhanced their market competitiveness. Relying on their strong consumer markets and logistics hub status, Shanghai and Beijing have become important circulation nodes for agricultural product e-commerce. With the support of national policies, central and western provinces such as Henan, Sichuan, and Hubei have improved logistics infrastructure, developed cold chain logistics, and improved the efficiency of e-commerce circulation of agricultural products. Anhui, Hunan and other countries have enhanced their market competitiveness through the construction of rural e-commerce infrastructure. Middle-ranking provinces such as Fujian, Tianjin and Chongqing have performed strongly thanks to their logistics hubs and specialty agricultural export markets. Due to geographical restrictions and insufficient logistics networks, the development of agricultural product e-commerce in western and northeastern regions such as Heilongjiang, Jilin, and Xinjiang is limited. In the future, through policy

**Table 26. Agricultural products e-commerce economic development capability weighted decision matrix X.**

| X | C 11 | C 12 |
|---|---|---|
| A 1 | 0.2828 | 0.1074 |
| A 2 | 0.1777 | 0.3273 |
| A 3 | 0.2764 | 0.1858 |
| A 4 | 0.0435 | 0.0000 |
| A 5 | 0.6006 | 0.1657 |
| A6 | 0.1063 | 0.0763 |
| A7 | 0.1058 | 0.0223 |
| A8 | 0.0210 | 0.1007 |
| A9 | 0.1767 | 0.1043 |
| A10 | 0.3221 | 0.0946 |
| A11 | 0.0610 | 0.0929 |
| A12 | 0.2916 | 0.1093 |
| A13 | 0.2492 | 0.1069 |
| A14 | 0.0419 | 0.0865 |
| A15 | 0.5716 | 0.2365 |
| A16 | 0.1650 | 0.1126 |
| A17 | 0.1197 | 0.1122 |
| A18 | 0.0577 | 0.1083 |
| A19 | 0.0083 | 0.0618 |
| A20 | 0.0016 | 0.0332 |
| A21 | 0.4422 | 0.1441 |
| A22 | 0.0930 | 0.0602 |
| A23 | 0.1316 | 0.0513 |
| A24 | 0.2138 | 0.3994 |
| A 25 | 0.3180 | 0.0943 |
| A 26 | 0.0387 | 0.2442 |
| A 27 | 0.0001 | 0.0876 |
| A 28 | 0.0342 | 0.0635 |
| A 29 | 0.1375 | 0.0432 |
| A 30 | 0.4045 | 0.3680 |
| A 31 | 0.1795 | 0.1036 |

support and technological innovation, these regions will improve their e-commerce infrastructure and unleash their agricultural products e-commerce economy through " rural revitalization " and the " Belt and Road " initiative.

## 5. Discussion

Heilongjiang Province has a significant gap in agricultural product e-commerce logistics service capabilities compared to coastal provinces, mainly due to lagging infrastructure construction, insufficient policy support, and relatively weak market dynamics. These factors have worked together to limit Heilongjiang Province's competitiveness in the field of agricultural product e-commerce logistics.

### 5.1. Lagging infrastructure construction

Logistics infrastructure is the core of the operation of the e-commerce logistics system, and Heilongjiang Province still has a large gap compared with coastal provinces in this regard.

**Table 27. Ideal value, negative ideal value and proximity of agricultural product e-commerce economic development capability.**

| province | Distance from positive ideal value$d^+$ | Distance from negative ideal value$d^-$ | Relative proximity C | Ranking |
|---|---|---|---|---|
| Anhui | 0.4316 | 0.3024 | 0.4120 | 11 |
| Beijing | 0.4290 | 0.3724 | 0.4647 | 6 |
| Fujian | 0.3882 | 0.3330 | 0.4617 | 7 |
| Gansu | 0.6855 | 0.0434 | 0.0596 | 30 |
| Guangdong | 0.2337 | 0.6230 | 0.7272 | 3 |
| Guangxi | 0.5905 | 0.1308 | 0.1813 | 20 |
| Guizhou | 0.6221 | 0.1081 | 0.1480 | 24 |
| Hainan | 0.6521 | 0.1028 | 0.1362 | 25 |
| Hebei | 0.5165 | 0.2052 | 0.2843 | 15 |
| Henan | 0.4128 | 0.3357 | 0.4484 | 8 |
| Heilongjiang | 0.6206 | 0.1111 | 0.1518 | 23 |
| Hubei | 0.4238 | 0.3113 | 0.4235 | 10 |
| Hunan | 0.4571 | 0.2711 | 0.3723 | 12 |
| Jilin | 0.6404 | 0.0960 | 0.1304 | 26 |
| Jiangsu | 0.1655 | 0.6185 | 0.7890 | 1 |
| Jiangxi | 0.5216 | 0.1996 | 0.2768 | 16 |
| Liaoning | 0.5601 | 0.1640 | 0.2265 | 17 |
| Inner Mongolia | 0.6160 | 0.1227 | 0.1661 | 21 |
| Ningxia | 0.6817 | 0.0623 | 0.0837 | 29 |
| Qinghai | 0.7021 | 0.0332 | 0.0451 | 31 |
| Shandong | 0.3004 | 0.4650 | 0.6075 | 4 |
| Shanxi | 0.6105 | 0.1107 | 0.1535 | 22 |
| Shaanxi | 0.5841 | 0.1412 | 0.1946 | 19 |
| Shanghai | 0.3869 | 0.4529 | 0.5393 | 5 |
| Sichuan | 0.4159 | 0.3316 | 0.4436 | 9 |
| Tianjin | 0.5829 | 0.2472 | 0.2978 | 13 |
| Tibet | 0.6767 | 0.0875 | 0.1145 | 27 |
| Xinjiang | 0.6585 | 0.0721 | 0.0987 | 28 |
| Yunnan | 0.5842 | 0.1441 | 0.1979 | 18 |
| Zhejiang | 0.1986 | 0.5468 | 0.7336 | 2 |
| Chongqing | 0.5146 | 0.2072 | 0.2871 | 14 |

**5.1.1. Traffic network restrictions.** Heilongjiang Province is located in the northeastern border of my country. Compared with the economically developed areas in the southeastern coastal areas, it is relatively far away from major consumer markets such as East China and South China, resulting in long logistics transportation cycles and high costs. Although the transportation infrastructure in the province has been continuously improved in recent years, and the highway, high-speed rail, and trunk railway networks have been gradually improved, such as the Harbin-Jiamusi Expressway and the Mudanjiang-Jiamusi High-speed Railway, it is still difficult to compete with logistics highlands such as the Yangtze River Delta and the Pearl River Delta in terms of overall transportation efficiency.

Especially in terms of cold chain logistics, Heilongjiang's cold chain transportation system is still in its early stages of development, and there is a phenomenon of "cold chain on the trunk line but lack of guarantee at the end". Fresh agricultural products often face the risk of "chain break" in the post-harvest storage and transportation links, which makes it more difficult to keep the products fresh and the loss rate is high. For example, when transporting specialty products such as

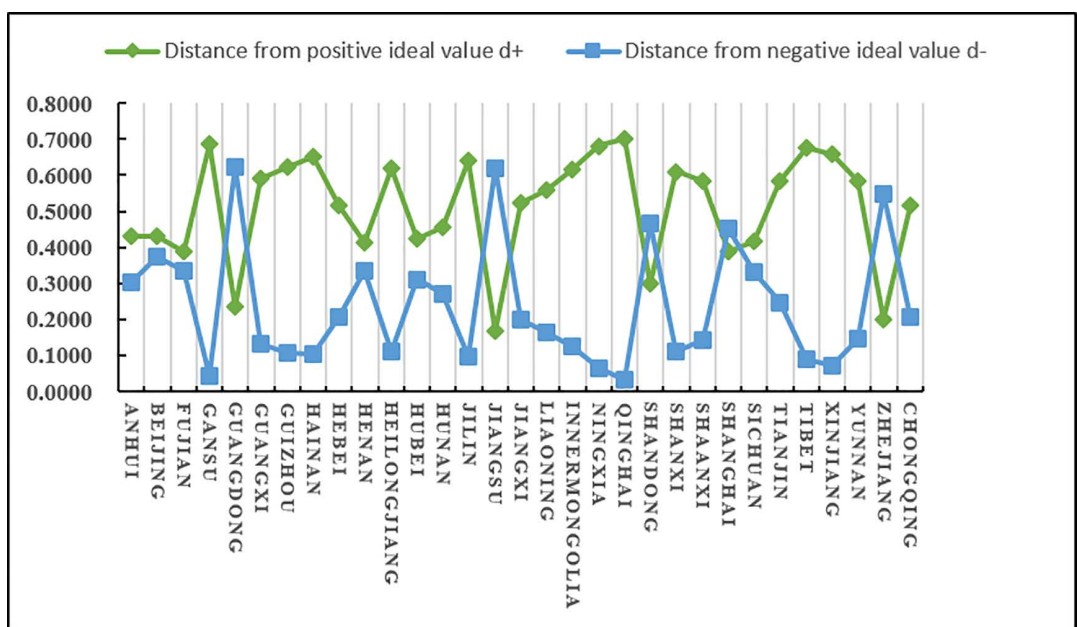

**Fig 12. The distance between the agricultural products e-commerce economic development capacity of each province and city and the positive and negative ideal values.**

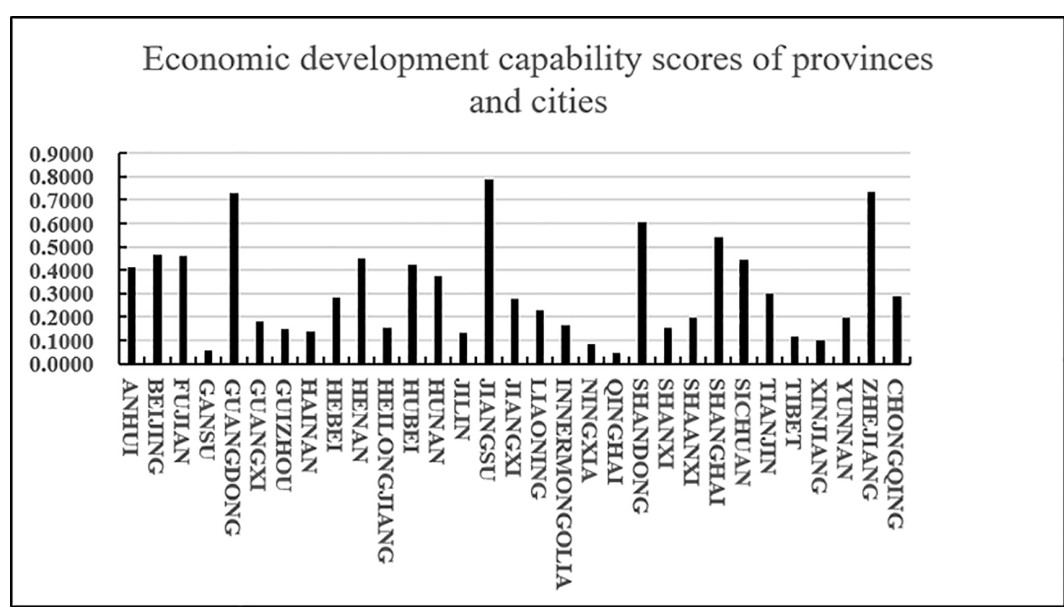

**Fig 13. Economic development capability scores of provinces and cities.**

blueberries and rice in high temperature seasons, if the temperature control cannot be achieved throughout the process, the quality is easily affected, which seriously restricts its commercialization level and market competitiveness.

In contrast, the Yangtze River Delta region has built a complete multimodal cold chain system based on its dense highway network, convenient aviation and port transportation, which can achieve efficient circulation of agricultural products

"picked on the same day and put on the shelves the next day". On this basis, it also relies on the big data dispatching system to optimize transportation routes and minimize energy consumption, and its logistics efficiency and economic advantages are significantly better than Heilongjiang.

**5.1.2. Weak warehousing and distribution system.** The development of agricultural product e-commerce has put forward higher requirements for warehousing, sorting and terminal distribution. Especially in the context of the gradual rise of the "forward warehouse + instant distribution" model, the modern warehousing and distribution system has become the core competitiveness of e-commerce logistics. However, Heilongjiang's layout in this regard is relatively lagging, and there are the following problems:

First, the storage facilities are old and the spatial layout is unreasonable. Most cold storages still use traditional models, lack constant temperature, automation, and information management systems, and cannot adapt to the high-frequency and small-batch order model of e-commerce platforms. Some county areas even still rely on manual operations, unable to achieve standardized sorting and rapid turnover.

Second, the intelligent sorting and terminal distribution capabilities are weak. In the Yangtze River Delta and other regions, intelligent warehousing and distribution technology is widely used in the sorting of agricultural products, including the use of visual recognition, automatic weighing, grading and classification equipment, which greatly improves the efficiency of order processing. However, most e-commerce companies in Heilongjiang are still in the "human workshop" operation stage, which is not only inefficient, but also prone to problems such as wrong delivery and missed delivery, affecting customer experience.

Third, the terminal distribution network coverage is incomplete. The express delivery coverage rate in some remote areas of Heilongjiang is not high, and the "last mile" delivery relies on rural couriers and self-pickup points. Problems such as untimely delivery and difficulty in returning and exchanging goods frequently occur, which significantly reduces the overall logistics service quality.

In contrast, the Yangtze River Delta and Pearl River Delta regions have formed a three-level warehousing layout of "city warehouse + community warehouse + cold chain terminal", combined with an intelligent dispatching system to achieve precise delivery, effectively control logistics costs and ensure service timeliness, and enhance stickiness to consumers, especially fresh food users.

## 5.2. Relative lack of policy support

The policy environment plays an important guiding role in the development of e-commerce logistics, but Heilongjiang's policy support in this area is relatively weak.

**5.2.1. Traffic network restrictions.** Although Heilongjiang has rich resource endowments and industrial foundations in developing agricultural product e-commerce logistics, there is still a clear gap in policy support compared with developed coastal provinces. Coastal provinces such as Zhejiang and Guangdong have generally established special agricultural product e-commerce support policy systems, covering:

Special fund subsidies: used to support agricultural product upstream, e-commerce platform construction, cold chain transportation, storage facility renovation, etc.;

Infrastructure investment: through the guidance of fiscal funds, accelerate the construction of modern logistics parks, smart cold chain hubs and other key nodes;

Cold chain transportation support policies: provide freight subsidies, cold chain vehicle purchase discounts, operation subsidies, etc. for fresh agricultural product transportation to reduce circulation costs.

Taking Zhejiang as an example, its "E-commerce into Rural Comprehensive Demonstration Project" has driven a large number of specialty agricultural products to achieve rapid circulation through e-commerce channels, and e-commerce and logistics companies have formed a policy-driven benign interaction mechanism. In contrast, Heilongjiang Province's current agricultural product e-commerce logistics support policies are relatively scattered, and some local governments

have the orientation of "focusing on industry and neglecting circulation". Policy incentives are still in the exploration and pilot stage, lacking systematicity and continuity, resulting in slow development of agricultural product e-commerce logistics chains and insufficient enthusiasm for social capital investment.

In addition, some support measures have problems such as complicated application procedures, long approval cycles, and slow disbursement of funds during implementation, making it difficult for policy dividends to truly take effect.

**5.2.2. Limited leading role of local governments.** In the development of modern e-commerce logistics system, the guiding and coordinating role of the government is crucial. The governments of developed coastal areas have shown stronger organizational ability and foresight in promoting the development of e-commerce logistics, including:

Promoting the development of industrial clusters: For example, Foshan, Guangdong, Yiwu, Zhejiang, etc., relying on local industrial foundations, build agricultural product e-commerce cluster parks, gather upstream and downstream resources such as processing, packaging, warehousing, logistics, and live sales to form an ecological closed loop;

Optimizing supply chain management: By introducing big data platforms, establishing supply chain financial mechanisms, and integrating farmers and market information, the intelligent, flat, and efficient production and marketing docking can be achieved;

Smart logistics construction: vigorously promote new smart logistics technologies such as "digital warehousing", "smart distribution", and "unmanned distribution", and provide support policies in finance, data, land, etc. to promote the digital transformation of traditional logistics.

However, the government role in Heilongjiang in this regard is still insufficient. On the one hand, there is a lack of top-level design and overall planning in the layout of e-commerce logistics space and the design of logistics channels; on the other hand, the promotion of smart logistics pilots and demonstration project cultivation is slow, and the digital transformation mechanism led by local governments has not yet been established. This not only affects the overall improvement of logistics service levels, but also weakens the support foundation for the development of agricultural product e-commerce. In addition, there is weak coordination among different regions in Heilongjiang Province in the government's promotion of e-commerce logistics development, and there is a lack of a provincial-level resource integration platform, making it difficult to form an overall synergy for regional linkage.

## 5.3 Gap between market demand and competitive environment

Market demand and industry competition environment directly affect logistics service capabilities, and Heilongjiang also faces certain challenges in this regard.

**5.3.1. Insufficient market demand.** The development of e-commerce logistics is highly dependent on the consumption capacity and consumption habits of the local market. In developed coastal areas such as Guangdong, Zhejiang, and Jiangsu, the population is dense and the residents have high income levels, and the e-commerce penetration rate is always at the leading level in the country. Data show that in 2022, Zhejiang's rural online retail sales have exceeded 400 billion yuan, accounting for an important share of the national rural e-commerce market, which directly drives the high-frequency distribution and network optimization of logistics companies, forming a virtuous cycle of "high demand-high investment-high efficiency".

In contrast, Heilongjiang Province has a small total population and a relatively scattered distribution. The urbanization rate and per capita consumption capacity are relatively low, especially in rural areas, where the online retail penetration rate is still low. Some remote areas are limited by high logistics costs, weak e-commerce awareness, and imperfect payment and delivery systems, and e-commerce consumption habits have not yet been fully formed. This phenomenon of "weak demand" on the consumer side makes it difficult for e-commerce logistics companies to dilute costs through large-scale operations, thereby suppressing their enthusiasm for warehousing construction, distribution coverage, and technology investment.

In addition, the relatively low frequency of orders makes it difficult for logistics companies to implement the application of key technologies such as intelligent scheduling and warehouse network optimization, which further leads to slow improvements in logistics efficiency and service levels, and falls into a vicious cycle of "small market - low investment - weak service - smaller demand", forming a systematic development gap with the developed coastal areas.

**5.3.2. The agglomeration effect of enterprises is not obvious.** Industrial agglomeration is an important driving force for the efficient operation of e-commerce logistics. In coastal cities such as Yiwu, Zhejiang and Shenzhen, Guangdong, by introducing leading e-commerce and logistics companies such as Alibaba, JD.com, and SF Express, a complete ecological system with coordinated development of platforms, e-commerce, warehousing, sorting, distribution, cold chain, and finance has been formed. This development model led by leading enterprises and linked upstream and downstream not only improves the level of regional logistics services, but also promotes the deep integration of information flow, capital flow and logistics.

In Heilongjiang, on the other hand, e-commerce logistics companies generally have the characteristics of "small, scattered, and weak", with limited layout of large platforms and third-party logistics companies, and a lack of leading companies with industry-leading roles in the local area. Most companies are in the primary stage of consignment and warehousing and distribution integration, and it is difficult to attract capital and technology investment, resulting in fragmented industrial chains, weak coordination capabilities, narrow logistics network coverage, single service types, and low overall operating efficiency.

More importantly, the lack of enterprise agglomeration also means insufficient innovation vitality and resource integration capabilities. Relying on the agglomeration effect, coastal provinces can quickly incubate new business forms such as smart logistics enterprises, logistics financial platforms, and logistics technology research and development institutions, while Heilongjiang faces the development bottleneck of "backward technology, weak service capabilities, and poor customer stickiness". In addition, the coordination mechanism and supply chain collaboration model between enterprises in the agglomeration area have not been established, which restricts industrial integration and value enhancement.

In summary, the development of Heilongjiang's agricultural products e-commerce logistics is faced with multiple overlapping challenges such as "location constraints + weak foundation + policy absence + insufficient market + weak entities". To achieve breakthrough development, it is necessary to coordinate and promote from aspects such as optimizing infrastructure, strengthening policy guidance, stimulating market potential, cultivating leading enterprises, and improving digital capabilities, and gradually build an agricultural products e-commerce logistics system with regional characteristics, efficient services, and integrated factors, so as to promote Heilongjiang's high-quality agricultural products to enter the national and even international markets faster and better.

# 6. Research conclusions and countermeasures

## 6.1. Research conclusions

This study evaluates the agricultural e-commerce logistics service capabilities of Heilongjiang Province from 2013 to 2022, focusing on five dimensions: logistics carrying capacity, technological innovation, operational efficiency, economic support, and national benchmarking. The key findings are as follows:

**6.1.1. Logistics carrying capacity remains insufficient, with notable regional disparities.** Despite progress in expressways and trunk railways, the overall logistics network density and transportation efficiency in Heilongjiang lag behind coastal regions like Guangdong and Jiangsu. Rural areas face significant challenges in "last-mile" delivery, and the cold chain system remains incomplete—especially at the terminal level—resulting in high product loss and compromised quality. Strengthening trunk-to-terminal connectivity and expanding cold chain infrastructure are essential steps forward.

**6.1.2. Technological innovation is underway but uneven, with limited intelligence.** Some areas have adopted digital logistics tracking systems, enhancing transparency and consumer trust. However, the province overall suffers from low levels of intelligence in logistics operations, especially in rural zones with poor information infrastructure. Compared

with more advanced provinces, Heilongjiang lacks digital logistics platforms and intelligent warehousing solutions. Investments in smart logistics infrastructure and data-driven management are urgently needed.

**6.1.3. Operational capacity has grown, yet supply chain coordination is lacking.** While logistics throughput and market demand have increased, most enterprises are still limited to basic warehousing and distribution services. Regional coordination remains weak, resulting in high operating costs and inefficient order fulfillment. Unlike the integrated logistics models of coastal provinces, Heilongjiang's logistics landscape is fragmented and in need of unified platforms and collaborative networks.

**6.1. 4. Economic growth supports logistics, but regional gaps persist.** Rising income levels and rural e-commerce growth have driven logistics expansion. However, economic vitality remains lower than in eastern regions, limiting infrastructure investment and service innovation, especially in underdeveloped rural areas. To boost logistics development, rural economic potential must be unlocked, and targeted support provided in high-priority regions.

**6.1.5. Comparative national performance highlights structural shortcomings.** Heilongjiang still trails provinces like Zhejiang, Jiangsu, and Guangdong in logistics integration, cold chain coverage, digital transformation, and policy support. High costs, delayed service response, and infrastructure deficits hinder competitiveness. Bridging this gap requires systemic upgrades in infrastructure, technology, and government policy to shift from "growth in volume" to "improvement in quality."

## 6.2. Recommendations

**6.2.1. Strengthen logistics infrastructure.** It is recommended to increase investment in rural transportation, introduce social capital to improve the road network, and upgrade cold chain logistics facilities. Support logistics companies to expand transportation scale through subsidies and tax incentives. Optimize "last mile" delivery, improve postal outlets, and promote collaboration with e-commerce and logistics platforms. Develop intelligent logistics and improve the level of informatization.

**6.2.2. Promoting technological innovation capabilities.** It is recommended to increase the construction of rural information infrastructure, promote 5G coverage, and encourage enterprises to participate. Promote the application of digital technologies such as blockchain to improve the transparency and efficiency of agricultural product e-commerce. Support the introduction of technologies such as smart warehousing, set up innovation funds, cultivate logistics talents, and promote the development of intelligent logistics.

**6.2.3. Optimize logistics operation management.** It is recommended to promote the application of digital technology in logistics enterprises to improve response speed and efficiency. Strengthen cooperation with leading logistics and e-commerce platforms to optimize network layout. Support technology upgrades for small and medium-sized logistics enterprises, provide tax incentives and loans. Build regional logistics centers, develop intelligent delivery such as drones, and solve the "last mile" problem in remote areas.

**6.2.4. Promote rural economic development.** It is recommended to set up a special e-commerce fund to support the brand building and promotion of agricultural enterprises and increase the added value of agricultural products. Promote the "e-commerce + agriculture" model, expand the market, and provide logistics support. Carry out e-commerce training to improve farmers' operational capabilities. Strengthen the construction of quality supervision and traceability systems, enhance consumer trust, and promote the development of agricultural product e-commerce.

**6.2.5. Comprehensive suggestions for improving the e-commerce logistics service capabilities of agricultural products in Heilongjiang Province.** It is recommended to increase the construction of rural roads and cold chain logistics, promote the PPP model, and develop multimodal transport. Promote the digitization and intelligence of logistics, and apply technologies such as the Internet of Things and big data. Introduce modern supply chain management systems to optimize warehousing and distribution. Strengthen brand building and e-commerce training to enhance the market competitiveness of agricultural products.

### 6.3. Limitations and future research

**Methodological limitations.** This study employs the entropy weight-TOPSIS method, offering objectivity and clarity but lacking in subjectivity and dynamic analysis. It assumes indicator independence, which overlooks interrelationships such as between logistics capacity and operations. Moreover, reliance on government data may limit precision due to reporting delays and regional inconsistencies.

**Future research directions.** Future work should integrate structural equation modeling (SEM), spatial econometrics, and dynamic panel data analysis to capture causal relationships and time-based trends. Combining subjective weighting techniques and machine learning can improve model adaptability and accuracy, providing stronger guidance for logistics system optimization (Fig 14).

## Appendix

### Appendix 1

Agricultural products e-commerce logistics services are the development path of the logistics industry from "traditional logistics" to "modern logistics", and its connotation is very rich. This article selects 12 indicators from four dimensions: logistics carrying capacity, technological innovation capability, operational capability, and economic development capability to construct an evaluation index system for Heilongjiang agricultural products e-commerce logistics service capabilities. The index system is shown in Table 2.1.

Logistics carrying capacity: The construction of logistics carrying capacity is one of the key factors in the logistics service capacity of agricultural products e-commerce. This indicator reflects the development of agricultural products logistics

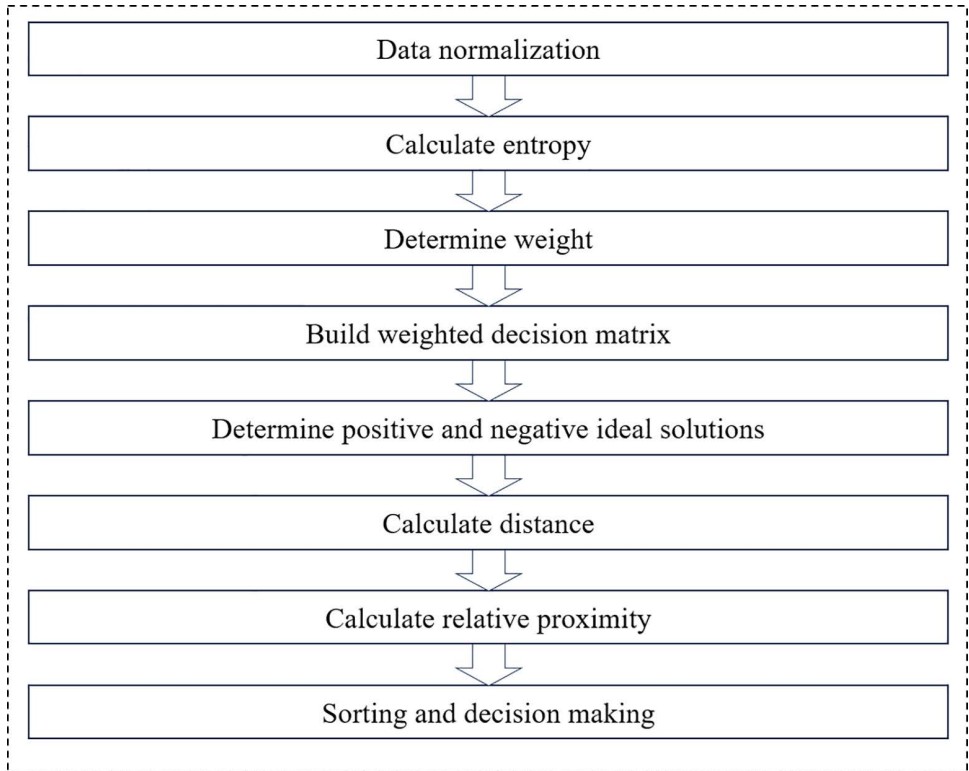

**Fig 14. Entropy weight-TOPSIS method flow chart.**

**Table 2. 1. Evaluation index system of Heilongjiang agricultural products e-commerce logistics service capabilities.**

| Target layer | Indicator layer | Indicator properties | vari-able |
|---|---|---|---|
| Logistics carrying capacity B1 | Mileage of expressways (km) | Positive | C1 |
| | Number of highway-operated trucks (10,000 vehicles) | Positive | C2 |
| | Postal business outlets | Positive | C3 |
| Technological innovation capability B2 | Mobile Internet users (10,000 households) | Positive | C4 |
| | Average number of mobile phones per 100 rural households at the end of the year (units) | Positive | C5 |
| Operational Capability B3 | Express delivery volume (10,000 pieces) | Positive | C6 |
| | Agricultural product logistics turnover (100 million ton-kilometers) | Positive | C7 |
| | Agricultural product logistics costs (trillion) | Positive | C8 |
| | Total agricultural product logistics (trillion) | Positive | C9 |
| | Added value of transportation, warehousing and postal services (100 million yuan) | Positive | C10 |
| Economic Development Capability B4 | Total retail sales of consumer goods (100 million yuan) | Positive | C11 |
| | Per capita disposable income of rural residents (yuan) | Positive | C12 |

carrying capacity in Heilongjiang Province. It includes the following indicators: mileage of expressways, number of road-operated trucks, and postal business outlets. The development of the highway network makes it easier to transport agricultural products from one region to another, and even covers more distant urban and rural markets. The number of trucks directly affects the scale and capacity of logistics transportation. More trucks mean greater transportation capacity, which can meet the needs of large-scale distribution of agricultural products e-commerce. The increase in postal business outlets can cover more areas, especially remote areas, which will help agricultural products e-commerce extend its services to a wider consumer market.

Technological innovation capability: Technological innovation capability plays an important role in the logistics service capability of agricultural products e-commerce. The popularity of mobile phones in rural areas has promoted the development of rural e-commerce, helping farmers to sell agricultural products more conveniently, and also allowing logistics services to be extended to rural areas. The Internet makes agricultural product logistics information more transparent, and consumers can more easily understand the production process and logistics status of agricultural products, which enhances consumer trust. Therefore, mobile Internet users and rural residents owning mobile phones are selected as indicators.

Operational capability: Improving logistics operational capability is of great significance to the development of logistics services. Specifically, improving logistics operational capability is of great significance to improving the competitiveness of logistics enterprises, optimizing supply chain efficiency, and promoting logistics service capabilities. This indicator reflects the operational capability of agricultural product e-commerce in Heilongjiang Province. It includes express delivery volume, agricultural product logistics turnover, agricultural product logistics costs, total agricultural product logistics, transportation, warehousing and postal business added value.

Economic development capability: Economic development capability plays an important supporting role in logistics service capability. It mainly includes the total retail sales of consumer goods and the per capita disposable income of rural residents. The increase in total retail sales means an increase in demand. Logistics companies can reduce unit transportation costs and improve service efficiency through economies of scale, thereby improving logistics service capabilities. If per capita income increases, in order to meet higher service requirements, logistics companies will optimize distribution networks and efficiency and improve overall service capabilities .

**Appendix 2**

**Appendix 3**

**Basic principles of entropy weight method.** The entropy weight method is a commonly used method for determining the weights of multiple indicators, which is used to perform weighted analysis on indicators [10]. It is based on the concept of information entropy and reflects the uncertainty and information content of indicators by calculating the entropy value of indicators. In the entropy weight method, the weight of an indicator is inversely proportional to its entropy value, that is, the larger the entropy value, the smaller the indicator weight.

The steps of the entropy weight method are as follows:

(1) Collect data for each evaluation indicator.

(2) Calculate the normalized value of each indicator and convert the indicator data into dimensionless form.

(3) Calculate the entropy value of each indicator. According to the distribution of indicator data, the larger the entropy value, the higher the uncertainty of the indicator.

(4) Calculate the weight of each indicator, normalize the entropy value of the indicator and take its relative value as the weight of the indicator.

(5) Normalize the weights of each indicator to ensure that the sum of the weights is 1.

The entropy weight method can effectively determine the weight of the indicator, so that the evaluation indicator can more accurately reflect the actual situation and improve the scientificity and reliability of decision-making. The entropy weight method is widely used in evaluation and analysis in various fields, including economics, management, environment and so on.

The basic steps of entropy weight calculation are as follows:

(1) Constructing a data matrix

Assume that the object to be evaluated is a development status with n evaluation objects and m evaluation indicators. The original data can be used to construct an evaluation matrix to form a data matrix of the original indicators.

$$A = \begin{pmatrix} X_{11} & \cdots & X_{1n} \\ \vdots & \ddots & \vdots \\ X_{n1} & \cdots & X_{nm} \end{pmatrix}_{n \times m} \tag{4-1}$$

Where X_ijis the value of the indicator iof the scheme j.

(2) Standardization of raw data:

$$\text{Positive indicators}: \ y_{ij} = \frac{x_{ij} - \min(x_j)}{\max(x_j) - \min(x_j)} \tag{4-2}$$

$$\text{Negative indicator}: \ y_{ij} = \frac{\max(x_j) - x_{ij}}{\max(x_j) - \min(x_j)} \tag{4-3}$$

Where $y_{ij}$ represents the value after data standardization, $x_{ij}$ represents the original value of the indicator in region $i$j.

(3) Calculate $j$ the proportion of the indicator value $P_{ij}$ in the year under the indicator $i$:

$$P_{ij} = \frac{y_{ij}}{\sum_{i=1}^{n} y_{ij}}$$

(4–4)

(4) Entropy calculation:

$$e(y_i) = -\sum_{i=1}^{m} (y_{ij} \ln y_{ij})$$

(4–5)

$$e_j = \frac{e(y_j)}{\ln m}$$

(4–6)

$$d_j = 1 - e_j$$

(4–7)

In the formula, m represents the research object, n represents the research index $j = 1, 2, \cdots m$, $i = 1, 2, \cdots n$

(5) Entropy value solution:

$$w_j = \frac{d_j}{\sum_{i=1}^{n} d_i}$$

(4–8)

In the formula $j = 1, 2, \cdots m$, $i = 1, 2, \cdots n$

(6) Calculate the comprehensive score:

$$F_i = \sum_{j=1}^{m} w_j p_{ij}$$

(4–9)

In the formula $j = 1, 2, \cdots m$, $i = 1, 2, \cdots n$

## Supporting information

**S1 Raw Data. Raw data used in the evaluation of agricultural products e-commerce logistics service capabilities.** (XLSX)

## Author contributions

**Conceptualization:** huan chen.

**Data curation:** huan chen.

**Funding acquisition:** Wei chen.

**Investigation:** Wei chen.

**Methodology:** Wei chen.

**Project administration:** Wei chen.

**Supervision:** Jianhui Yin, Weinan Sun.

**Validation:** Jianhui Yin, Weinan Sun.

**Visualization:** huan chen, Jianhui Yin, Weinan Sun.

**Writing – original draft:** huan chen.

**Writing – review & editing:** huan chen.

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
