## [Decision Letter · Decision Letter 0]

PONE-D-24-52199Evaluating Agro E-commerce Logistics Services in Heilongjiang via Entropy Weight TOPSISPLOS ONE

Dear Dr. chen,

Thank you for submitting your manuscript to PLOS ONE. After careful consideration, we feel that it has merit but does not fully meet PLOS ONE’s publication criteria as it currently stands. Therefore, we invite you to submit a revised version of the manuscript that addresses the points raised during the review process.

We look forward to receiving your revised manuscript.

Kind regards,

Bikash Koli Dey

Guest Editor

PLOS ONE

Journal Requirements:

1. Key project of Heilongjiang Province's economic and social development "Research on the quality and safety supervision mechanism and implementation path of high-quality agricultural products driven by digital economy" (Project No.: 23301);

2. Harbin Science and Technology Bureau Science and Technology Plan Project (Project No. ZC2023ZJ014007) "Research on the efficiency evaluation of green technology innovation in equipment manufacturing industry for intelligent manufacturing";

3. Heilongjiang Oriental College Scientific Research Innovation Team Construction Project (Project No.: HDFKYTD202108);

4. National Social Science Fund Project "Research on the identification and breakthrough of key core technologies in the IC industry from the perspective of "dual chain integration"" (23BGL076).

4. For studies involving third-party data, we encourage authors to share any data specific to their analyses that they can legally distribute. PLOS recognizes, however, that authors may be using third-party data they do not have the rights to share. When third-party data cannot be publicly shared, authors must provide all information necessary for interested researchers to apply to gain access to the data. (https://journals.plos.org/plosone/s/data-availability#loc-acceptable-data-access-restrictions) 

5. Please amend either the title on the online submission form (via Edit Submission) or the title in the manuscript so that they are identical.

6. Please upload a new copy of Figure xxxx as the detail is not clear. Please follow the link for more information: ""https://blogs.plos.org/plos/2019/06/looking-good-tips-for-creating-your-plos-figures-graphics/"" ""https://blogs.plos.org/plos/2019/06/looking-good-tips-for-creating-your-plos-figures-graphics/""

7.We note you have included a table to which you do not refer in the text of your manuscript. Please ensure that you refer to Table 4.3, 4.7, 4.11, and 4.15,in your text; if accepted, production will need this reference to link the reader to the Table.

Additional Editor Comments :

Your manuscript has been reviewed with the assistance of referees. Copies of their comments are enclosed. You will note that they are reluctant to recommend publication at this time. Suggestions have been made for certain modifications that the referees feel will improve the paper. Along with the concerns raised by the referee, please discussed the following queries:

The additional considerations regarding the implementation challenges, such as training, costs, and technology adoption among local businesses, should be addressed. The paper should evaluate the potential impact of the policies on various stakeholders, including small farmers and businesses. A detailed risk assessment is required to understand how dependent the region is on external sources and what contingencies can be put in place to mitigate potential disruptions. Provide some references from this journal to ensure your paper within the scope of this journal.

Reviewers' comments:

Reviewer's Responses to Questions

**Comments to the Author**

1. Is the manuscript technically sound, and do the data support the conclusions?

Reviewer #1: Partly

Reviewer #2: Yes

2. Has the statistical analysis been performed appropriately and rigorously? 

Reviewer #1: Yes

Reviewer #2: Yes

3. Have the authors made all data underlying the findings in their manuscript fully available?

Reviewer #1: Yes

Reviewer #2: Yes

4. Is the manuscript presented in an intelligible fashion and written in standard English?

Reviewer #1: Yes

Reviewer #2: Yes

5. Review Comments to the Author

Reviewer #1: Authors analyze the characteristics of agricultural products e-commerce logistics services and builds an index system of agricultural products e-commerce logistics capabilities in Heilongjiang Province. The area of research seems interesting. However, the following aspects should be considered:

1. The precise research gap(s) and the major outcomes should be mentioned in the Abstract. Keywords need to be updated with standard keywords.

2. The Introduction needs to be revised while specifying the research questions. Mentioning a number of articles without proper interlinks make this paper less interesting for readers.

3. Citation of multiple references must be avoided.

4. Sect 2 describes Operational capability, Logistics carrying capacity etc., the majority of which consists of existing knowledge. Thus, these can be moved to appendix along with proper referencing.

5. Sect 3 should be improved with more details. Also, Sect 4.1.1 can be moved to appendix.

6. What is the source of data? Has the author followed the ethical practices? If yes, the ethical disclosure certificate must be enclosed as a supplemantary material.

7. Authors need to provide further analysis, in addition to those in Table 4.19. Describe on calculation of entropy value e and information utility value d of each item.

8. There are some typos and language quality should be improved.

Reviewer #2: The manuscript explores the logistics service capabilities for agricultural e-commerce in Heilongjiang Province using the entropy weight-TOPSIS method. It builds a multi-dimensional evaluation index system and provides a decade-long assessment of logistical developments while benchmarking against other Chinese provinces. The research highlights the increasing yet lagging performance of Heilongjiang's logistics services and proposes solutions for improvement.

The topic is highly relevant to the current focus on rural e-commerce and logistics optimization in China.

The combination of the entropy weight and TOPSIS method adds rigor and a systematic approach to evaluating logistics capabilities.

The manuscript employs detailed, decade-spanning data and statistical evaluations to support its conclusions.

Minor Comments and Suggestions:

1. The explanation of the entropy weight-TOPSIS method, while detailed, can benefit from a simplified flow diagram or visual representation to enhance clarity for broader audiences.

2. The manuscript mentions the use of SPSS to handle missing data but lacks details about the imputation method used. Please elaborate on how the missing values were filled and their potential impact on the results.

3. The discussion section could expand on why Heilongjiang underperforms relative to coastal provinces. Highlight specific challenges such as infrastructure, policies, or market dynamics.

4. The countermeasures suggested in the conclusion are general. Adding concrete steps, like specific government initiatives or partnerships, would strengthen the practical applicability of the recommendations.

5. Some figures (e.g., Figure 4.5) lack detailed labels or legends, making it hard to interpret them without referring to the text. Consider enhancing their readability.

6. Ensure references to studies from 2024 align with the journal's publication timeline. Future-dated references may need clarification or replacement with alternatives. Add the related article in references: K. Gao, T. Liu, Y. Rong et al., A novel BWM-Entropy-COPRAS group decision framework with spherical fuzzy information for digital supply chain partner selection, Complex & Intelligent Systems, 10 (2024), 6983-7008.

M. Aggarwal, R. Krishankumar, K.S. Ravichandran et al., Assessing potential of organizations with fuzzy entropy, Operations Research Forum, 4 (2023), 11, https://doi.org/10.1007/s43069-022-00178-0

S Sen, L Sahoo, K Tiwary et al., Entropy weighted TOPSIS based cluster head selection in wireless sensor networks under uncertainty, Telecom, 4 (2023), 678-692.

A. Hussain, K. Ullah et al., Energy supplier selection by TOPSIS method based on multi-attribute decision-making by using novel idea of complex fuzzy rough information, Energy Strategy Reviews, 54 (2024), 101442, https://doi.org/10.1016/j.esr.2024.101442

Z. Ali, Z.A. Khan, et al., Frank-based TOPSIS methodology of development and operations challenges based on intuitionistic linguistic aggregation operators and their applications, IEEE Access, 12 (2024), 91605-91618

7. The abstract can be condensed slightly to focus more on key findings and their significance while trimming excessive methodological detail.

8. While the manuscript is generally well-written, minor grammatical corrections and phrasing adjustments are needed in sections like the introduction (e.g., repetitive use of "e-commerce logistics service capabilities").

With minor revisions addressing the points above, the manuscript would make a valuable contribution to the literature on agro e-commerce logistics. I recommend acceptance with minor revisions.

6. PLOS authors have the option to publish the peer review history of their article (what does this mean? ). If published, this will include your full peer review and any attached files.

**Do you want your identity to be public for this peer review?** For information about this choice, including consent withdrawal, please see our Privacy Policy .

Reviewer #1: **Yes: ** Arindam Garai, Ph.D.

Reviewer #2: No

---

## [Author Response · Author response to Decision Letter 1]

10 Mar 2025

We are honored to have the opportunity to submit the revised manuscript of the paper " Evaluation of E-commerce Logistics Service Capability of Agricultural Products in Heilongjiang Province Based on Entropy Weight TOPSIS Method" to PLOS for publication. We thank the editors and reviewers for their time and effort in providing feedback on our manuscript and for their insightful comments and valuable improvements on our paper. We have adopted most of the suggestions made by the reviewers. Please see below for a one-by-one response to the reviewers' comments and concerns. All section numbers refer to the revised manuscript file, with revision records attached .

---

## [Decision Letter · Decision Letter 1]

PONE-D-24-52199R1Evaluation of agricultural products e-commerce logistics service capabilities in Heilongjiang Province based on entropy weight TOPSIS methodPLOS ONE

Dear Dr. chen,

Thank you for submitting your manuscript to PLOS ONE. After careful consideration, we feel that it has merit but does not fully meet PLOS ONE’s publication criteria as it currently stands. Therefore, we invite you to submit a revised version of the manuscript that addresses the points raised during the review process.

We look forward to receiving your revised manuscript.

Kind regards,

Bikash Koli Dey

Guest Editor

PLOS ONE

Journal Requirements:

Additional Editor Comments (if provided):

Revised the manuscript based on the comments provided by the reviewer.

Reviewers' comments:

Reviewer's Responses to Questions

**Comments to the Author**

1. If the authors have adequately addressed your comments raised in a previous round of review and you feel that this manuscript is now acceptable for publication, you may indicate that here to bypass the “Comments to the Author” section, enter your conflict of interest statement in the “Confidential to Editor” section, and submit your "Accept" recommendation.

Reviewer #1: All comments have been addressed

Reviewer #2: All comments have been addressed

Reviewer #3: All comments have been addressed

Reviewer #4: All comments have been addressed

2. Is the manuscript technically sound, and do the data support the conclusions?

Reviewer #1: Yes

Reviewer #2: Yes

Reviewer #3: Yes

Reviewer #4: Yes

3. Has the statistical analysis been performed appropriately and rigorously? 

Reviewer #1: N/A

Reviewer #2: Yes

Reviewer #3: Yes

Reviewer #4: Yes

4. Have the authors made all data underlying the findings in their manuscript fully available?

Reviewer #1: Yes

Reviewer #2: Yes

Reviewer #3: Yes

Reviewer #4: Yes

5. Is the manuscript presented in an intelligible fashion and written in standard English?

Reviewer #1: Yes

Reviewer #2: Yes

Reviewer #3: Yes

Reviewer #4: Yes

6. Review Comments to the Author

Reviewer #1: Authors followed and addressed all concern properly. The manuscript can be accepted for publication.

Reviewer #2: The author has revised the article according to my comments. I recommend it for publication in your reputed journal.

Reviewer #3: The paper can be accepted for publication. I have checked and verified the paper, but I could not see my earlier comments. That's why I could not verify the earlier comments' responses.

Reviewer #4: 1. An abstract should address these questions: what are you trying to do, why, what you found and what is the significance of your findings. Rewrite and improve.

2. There are several studies already developed in the literature. What is the utility of your proposed study is better than previous.

3. Comparative discussion needs further explanation. Please do more work on it.

4. Please improve the conclusion section. Also, limitations in the developed approach should be discussed in the conclusions section.

5. The notations used should be rechecked.

6. The writing is recommended to be improved. The authors are suggested to proofread paper and restructuring of sentences are required for the entire manuscript.

7. PLOS authors have the option to publish the peer review history of their article (what does this mean? ). If published, this will include your full peer review and any attached files.

**Do you want your identity to be public for this peer review?** For information about this choice, including consent withdrawal, please see our Privacy Policy .

Reviewer #1: **Yes: ** Arindam Garai

Reviewer #2: No

Reviewer #3: **Yes: ** Prof. Biswajit Sarkar

Reviewer #4: No

---

## [Author Response · Author response to Decision Letter 2]

24 Apr 2025

We sincerely thank the editor and all reviewers for their time, thoughtful feedback, and constructive suggestions on our manuscript titled “Evaluation of Agricultural Products E-commerce Logistics Service Capabilities in Heilongjiang Province Based on Entropy Weight TOPSIS Method.” Based on the comments received, we have carefully revised the manuscript and made the following key improvements:

Abstract Revision: As suggested by Reviewer #4, we revised the abstract to clearly address the research objective, significance, findings, and policy implications. The updated version provides a more compact and structured summary of the study.

Literature Contribution Clarified: We provided a detailed comparative discussion highlighting how our study advances prior literature in terms of data depth, indicator design, methodological rigor, and regional applicability.

Expanded Comparative Analysis: In response to the request for more explanation, we substantially expanded our comparative discussion (Section 5), elaborating on the differences between Heilongjiang and developed regions in transportation infrastructure, warehousing, policy support, market dynamics, and enterprise clustering.

Conclusion Enhanced & Limitations Added: The conclusion section was strengthened with more focused summaries of findings by dimension. In addition, a subsection discussing methodological limitations and future research directions was added to address the reviewer’s concern.

Notation and Symbol Consistency: We conducted a thorough check and standardization of all mathematical symbols, variables, and abbreviations throughout the manuscript to ensure clarity and consistency.

Language and Structure Improvement: The entire manuscript was comprehensively proofread, sentence structures were optimized, and terminology was standardized. We also utilized professional English editing tools to improve fluency and academic expression.

We have annotated all revisions in the manuscript and provided detailed point-by-point responses for each reviewer comment. We hope the revised version meets the publication standards and sincerely appreciate your continued consideration of our work.

---

## [Editor Report · Decision Letter 2]

PONE-D-24-52199R2Evaluation of agricultural products e-commerce logistics service capabilities in Heilongjiang Province based on entropy weight TOPSIS methodPLOS ONE

Dear Dr. chen,

Thank you for submitting your manuscript to PLOS ONE. After careful consideration, we feel that it has merit but does not fully meet PLOS ONE’s publication criteria as it currently stands. Therefore, we invite you to submit a revised version of the manuscript that addresses the points raised during the review process.

We look forward to receiving your revised manuscript.

Kind regards,

Bikash Koli Dey

Guest Editor

PLOS ONE

Journal Requirements:

Additional Editor Comments:

Dear author, before further process please rectify the following concenrs:

The articles published in this journal should be written in English language. In some figure other languages are detected which make the confusion for the reader.

Try to restuctured the section 6 as it is very long. You may consider to make different section.

Either use journal format to make the manuscript or should be consistant in writing with same font and size throughout the manuscript.

---

## [Author Response · Author response to Decision Letter 3]

5 May 2025

We sincerely thank the Editor for the careful review and valuable suggestions that have helped us improve the quality and clarity of our manuscript. Below is a summary of the revisions made in response to the Editor's comments:

Language in Figures: All figures and tables previously containing non-English text (Table 4.6 to Table 4.14) have been translated into English. These changes are reflected on pages 24–37 of the revised manuscript.

Section 6 Restructuring: Section 6 has been carefully revised to reduce its length and enhance clarity. While it was not split into sub-sections, redundant content was removed and the section is now more concise. Revisions are detailed on pages 40–49.

Formatting Consistency: The entire manuscript has been reformatted to ensure consistency in font and size. Specifically, the manuscript now uses Times New Roman, font size 9, throughout.

Reference Review: We have carefully reviewed all references and confirm that none of them have been retracted. Therefore, no changes were necessary in this regard.

We appreciate the Editor’s thoughtful feedback and the opportunity to revise our manuscript accordingly. We hope that the revised version meets the expectations of the journal.

---

## [Decision Letter · Decision Letter 3]

Evaluation of agricultural products e-commerce logistics service capabilities in Heilongjiang Province based on entropy weight TOPSIS method

PONE-D-24-52199R3

Dear Dr. chen,

We’re pleased to inform you that your manuscript has been judged scientifically suitable for publication and will be formally accepted for publication once it meets all outstanding technical requirements.

Kind regards,

Bikash Koli Dey

Guest Editor

PLOS ONE

Additional Editor Comments (optional):

Reviewers' comments:

Reviewer's Responses to Questions

**Comments to the Author**

1. If the authors have adequately addressed your comments raised in a previous round of review and you feel that this manuscript is now acceptable for publication, you may indicate that here to bypass the “Comments to the Author” section, enter your conflict of interest statement in the “Confidential to Editor” section, and submit your "Accept" recommendation.

Reviewer #4: All comments have been addressed

2. Is the manuscript technically sound, and do the data support the conclusions?

Reviewer #4: Yes

3. Has the statistical analysis been performed appropriately and rigorously? 

Reviewer #4: Yes

4. Have the authors made all data underlying the findings in their manuscript fully available?

Reviewer #4: Yes

5. Is the manuscript presented in an intelligible fashion and written in standard English?

Reviewer #4: Yes

6. Review Comments to the Author

Reviewer #4: The author improve the work according to my comments. Therefore, I recommended for accepting the paper.

7. PLOS authors have the option to publish the peer review history of their article (what does this mean? ). If published, this will include your full peer review and any attached files.

**Do you want your identity to be public for this peer review?** For information about this choice, including consent withdrawal, please see our Privacy Policy .

Reviewer #4: No

---

## [Editor Report · Acceptance letter]

PONE-D-24-52199R3

PLOS ONE

Dear Dr. chen,

I'm pleased to inform you that your manuscript has been deemed suitable for publication in PLOS ONE. Congratulations! Your manuscript is now being handed over to our production team.

Kind regards,

on behalf of

Dr. Bikash Koli Dey

Guest Editor

PLOS ONE